# Bifunctional Electrocatalysts Materials for Non-Aqueous Li–Air Batteries

**Guanghui Yue \***, **Zheyu Hong, Yongji Xia, Tianlun Yang and Yuanhui Wu**

College of Materials, Xiamen University, Xiamen 361005, China
\* Correspondence: yuegh@xmu.edu.cn

**Abstract:** Rechargeable lithium–air batteries (LABs), particularly the nonaqueous form, are demonstrated as the next-generation energy conversion and storage equipment for many applications. The air cathode has been certified as one of the critical aspects to affect the full performance of the LABs. At present, the main challenge for the commercial application of air cathode is to exploit some new cathode catalysts with lower prices, higher efficiency, and better stability. In the last decade, tremendous efforts have been dedicated to developing new structure design and fabrication processes of the cathode materials to promote the full capability of the LABs. The recent research progress of bifunctional cathode catalysts for LABs, including the main improvement strategies and performance of cathode materials, is introduced in this paper. Besides, related technical challenges are analyzed, and possible resolving strategies for the challenges to develop the nonaqueous Li–air battery catalytic cathodes are elaborated on in this review.

**Keywords:** Li-air battery; electrocatalyst; carbon-based materials; transition metal chalcogenides



## 1. Introduction

With the continuous progress of productivity, the consumption of energy is also increasing daily, which promotes the development of our modern society. Nowadays, facing the fossil fuel-based energy drying up crisis, the energy system has been transformed into clean and sustainable energy with less carbon dioxide emissions and environmental pollution. It is crucial to exploit rechargeable batteries with higher capacity density and better energy efficiency to combat climate changes and improve energy security and sustainability. Among various rechargeable battery technologies, Li–air batteries (Li-O$_2$ batteries, LABs) have a super-high theoretical specific energy [1–6] of 11,000 Wh kg$^{-1}$ (even with very conservative estimates and considering all battery components of the Li–air battery, the theoretical specific energy is around 3000 Wh kg$^{-1}$), which is much higher than the Li-ion batteries (around 150~200 Wh kg$^{-1}$ [5]). Therefore, they have attracted widespread attention for their extensive potential applications in electric vehicles (EV), laptops, and so on. Figure 1 shows the structure of LABs.

However, due to the low efficiency and poor performance of air electrocatalysts, which directly lead to their low-rate performance and poor long-term stability, the practical application of LABs still confronts severe bottleneck problems. This indicates that the LABs are still in the infancy of their research.

It is well known that the properties of LABs greatly rely on the sluggish chemical reactions of the electrocatalytic cathode, the oxygen reduction reaction (ORR), and the oxygen evolution reaction (OER). Air-electrode electrocatalysts must be active for both the ORR during the battery discharging process and the OER during the battery recharging process to improve the electrochemical behavior of LABs. In other words, an effective bifunctional electrocatalyst to improve the ORR and OER performance must be furnished for the LABs system, with which the charge and discharge of LABs can be controlled efficiently and directly, and the power, energy density, and energy efficiency of the LABs

should be guaranteed. Recent research has shown that the Pt or Pt-based nanocomposites as the electrocatalysts have the best ORR performance, [7,8] but the OER performance is limited since the forming of oxide layer when they worked at high potential. In addition, the Ir or Ru and their based nanocomposites as the electrocatalysts indicated highly effective OER performance and restricted ORR activity [9,10]. Therefore, individual performance coupled with the overpriced and the limited reserve of the rare metals caused the most formidable challenge for the development and practical application of the LABs. Hence, it is necessary for people to develop a kind of cheap electrocatalyst that can work with the ORR and OER synchronously and efficiently.

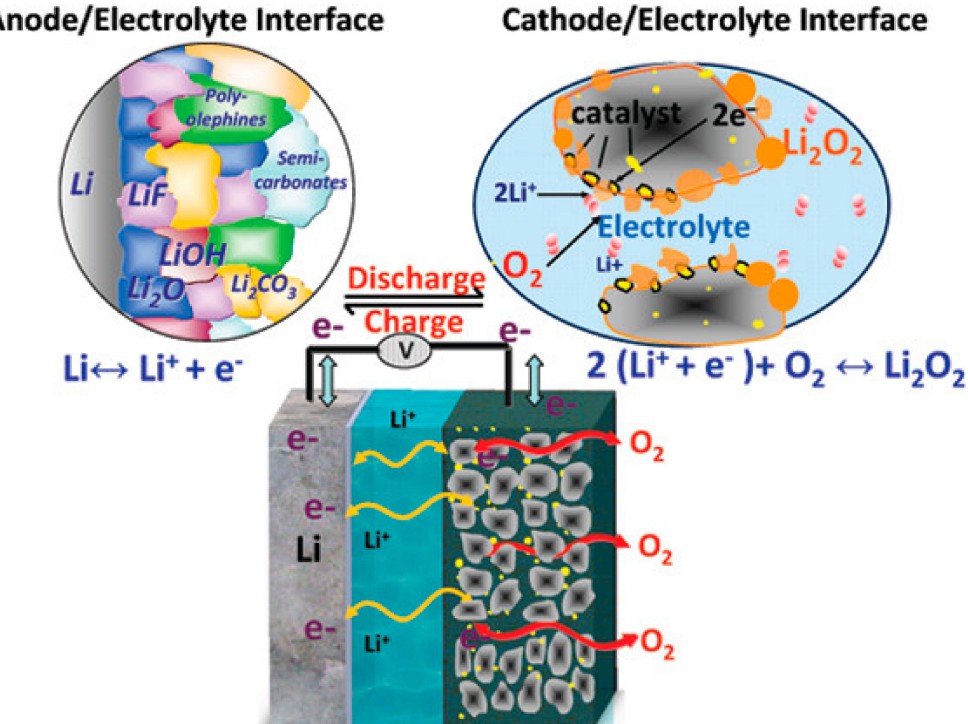

**Figure 1.** The architecture of LABs. Reprinted with permission from ref. [4]. Copyright 2010 American Chemical Society.

In this manuscript, electrocatalysts were classified into four different teams: noble or rare metals, non-noble transition metals/transition-metal-oxides, transition metals, and carbon-based materials. The critical essay bridges the gap between the antecedent reviews on the bifunctional (ORR/OER) catalysts and the LABs' performance. Moreover, the latest instance of the bifunctional electrocatalysts for rechargeable LABs has been displayed in this review.

## 2. Li–Air Batteries (LABs)

As mentioned above, as rechargeable batteries with higher theoretical energy density than the most advanced lithium-ion battery, LABs could be perceived as one of the greatest pieces of potential energy storage equipment in EV applications. However, there is still a long way before its use in practical applications. One of the main technical bottlenecks is developing cheap and highly efficient bifunctional cathode electrocatalysts for ORR/OER in the air cathodes. The typical aprotic LABs are made up of a lithium metal, a separator immersed with aprotic solvent electrolytes, and a porous carbon cathode. The ideal reactions of the non-aqueous LABs can be sketched below [5,11,12]:

Discharge

$$\text{Anode}: 2\text{Li} \rightarrow 2\text{Li}^+ + 2\text{e}^- \tag{1}$$

$$\text{Cathode}: \text{O}_2 + 2e^- + 2\text{Li}^+ \rightarrow \text{Li}_2\text{O}_2 \tag{2}$$

Charge

$$\text{Anode}: 2\text{Li}^+ + 2e^- \rightarrow 2\text{Li} \tag{3}$$

$$\text{Cathode}: \text{Li}_2\text{O}_2 \rightarrow \text{O}_2 + 2e^- + 2\text{Li}^+ \tag{4}$$

Overall reaction

$$2\text{Li} + \text{O}_2 \leftrightarrow \text{Li}_2\text{O}_2 \ E^0 = 2.96 \text{ V} \left(\text{vs. Li}^+/\text{Li}\right) \tag{5}$$

This means that in the discharge process, the metal lithium at the anode is oxidized to form $\text{Li}^+$, and the oxygen is charged into oxygen species (superoxide or peroxide) at the cathode, simultaneously. Then, the lithium ions are transported by the electrolyte and get to the cathode electrode to meet the reduced oxygen species forming the final product, an insoluble solid lithium peroxide ($\text{Li}_2\text{O}_2$), which will be absorbed on the surface of the porous substrate. With the following charge processing, the $\text{Li}_2\text{O}_2$ can be decomposed into oxygen and lithium ions. Then, the $\text{Li}^+$ will be transported back and reduced to form metal Li on the anode Li metal surface. Therefore, the electrochemistry of the LABs is dependent on the thermodynamics and the kinetics process of the $\text{Li}_2\text{O}_2$ formation. From the Equations (1)–(5) we can get that the standard free energy $\Delta_r G$ is around 570.18 kJ mol$^{-1}$ and the equilibrium potential is about 2.96 V (vs. Li$^+$/Li) for the overall chemical reaction. The super-high theoretical energy capacity, which is about 3500 Wh kg$^{-1}$ for the aprotic LABs, was calculated with the mass of the $\text{Li}_2\text{O}_2$. As an almost universal rule, electrochemical battery operation requires materials with good electrical conductivity, but the rechargeable product $\text{Li}_2\text{O}_2$ is a non-decomposable insulator material, and deposition on the cathode surface increases the difficulty of ORR/OER processing. Therefore, highly active materials are needed to improve the $\text{Li}_2\text{O}_2$ decomposition and enhance the cycle-life and efficiency of the LABs.

However, beyond this point, many complex by-products such as $\text{Li}_2\text{CO}_3$, $\text{HCO}_2\text{Li}$, $\text{CH}_3\text{CO}_2\text{Li}$, polyethers/polyesters, $\text{CO}_2$, and $\text{H}_2\text{O}$ can be produced by the electrolyte decomposition during the discharge/charge process [13–15]. Among them, the most destructive by-product is $\text{Li}_2\text{CO}_3$. Because it is one of the non-decomposable species, it will generate a large polarization during the charging process, reduce capacity retention, and greatly harm the cycle life and cycle stability of the LABs [16–18]. Therefore, highly active materials which can help the resolving of the $\text{Li}_2\text{CO}_3$ and the $\text{Li}_2\text{O}_2$ need to be developed to fulfill the imposed strict demands for oxygen batteries to match the high reactivity [19–21].

## 3. Bifunctional Catalysts for LABs Air Electrodes

Benefiting from the super-high theoretical energy density, LABs will be one of the potential techniques for energy storage and conversion devices. Many researchers have put forth their greatest efforts in scientifically researching the practical applications of the LABs. Although research on the LABs has been developed rapidly, there are still many problems to be solved, and there is still a large gap between the laboratory research and practical applications, including the poor cycle-life, low rate capacity, low energy efficiency, etc. [2,6,8,15,22–27]. However, how to improve the performance of the LABs is a very complex problem, such as using the suitable electrolyte, protecting the lithium anodes, promoting the stability of the cathode materials, and so on. Consequently, various aspects have become a popular research domain in the energy storage fields [28–31]. Chief among them is the development and application of the electrocatalysts, which are essential in the LABs field.

It is well known that the main product $\text{Li}_2\text{O}_2$ will be formed and decomposed on the surface of the cathode materials during the cycling of LABs. In the meanwhile, it is inevitable that higher overpotential will be detected instantly, which originated from the dull kinetics of the ORR and OER [32–37]. The higher overpotential in the charging process of the LABs will lead to poor energy efficiency, causing electrolyte decomposition and

electrode materials degradation. Therefore, exploring new efficient bifunctional catalysts to reduce overpotentials and accelerate reaction kinetics is an urgent and realistic "bottleneck" problem. We found that various carbon materials [33,35], noble metals [1,2,7], transition metals and their oxides/sulfides/nitride, [1,2,9,24,26,32] etc. have been used to enhance the ORR and/or OER. In terms of seduction, the various bifunctional catalysts will be discussed separately, and these research results will be served as a reference for future study.

### 3.1. Carbon Materials

With their ideal electrical conductivity and large surface area, carbon materials have been applied abroad in a variety of the energy storage systems, such as electrode materials, catalyst support, conductive agent, and so on. In the field of LABs, it has been used as the cathode catalyst in many studies. Carbon materials mainly include commercial porous carbon powders, carbon nanotubes (CNT), graphene, etc.

### 3.1.1. Carbon Powders

The super P carbon and the Ketjen black (KB) are the most common commercial carbon blacks. Research has indicated that the pore size and volume of the carbon powders show a great effect on the discharge capacity of the LABs compared to the surface area of the carbon powder [38–42]. Liu's group [42] has study about eight different kinds of carbon electrodes with a diverse variety of pore sizes and pore cubage. Their report displayed that the super P carbon with a low surface area of about 62 $m^2$ $g^{-1}$, but an activated carbon with a higher surface area of about 1200 $m^2$ $g^{-1}$ indicated a capacity of about 1900 mA h $g^{-1}$, which is much lower than the super P. A physical model was used to explain the influence of carbon pore size on battery capacity and the evolution of the $Li_2O_2$ (Figure 2b). Stability and chemical reaction studies revealed that the carbon is relatively stable when the discharge/charge potential is lower than 3.5 V (vs Li/Li+), but with the charge potential higher than 3.5 V, the irreversible by-product $Li_2CO_3$ could be detected, and lead to an unstable process (Figure 2a) [26].

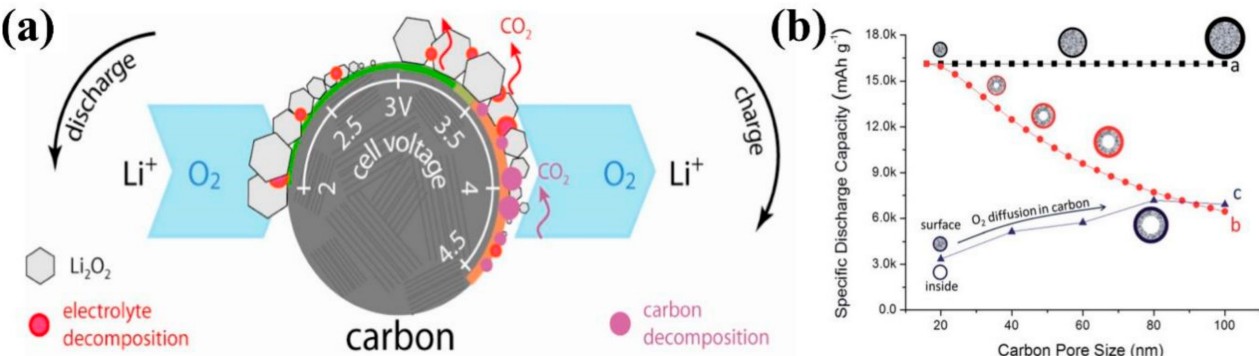

**Figure 2.** (**a**) Scheme for the decomposition of carbon electrode in LABs. Reprinted with permission from ref. [26]. Copyright 2013 American Chemical Society. (**b**) Physical model of $Li_2O_2$ in porous carbon: Line a indicates that all carbon pores are filled with $Li_2O_2$; Line b assumes that a $Li_2O_2$ monolayer (with a thickness of 7.8 nm) is formed in the carbon pores; Line c is the experimental data. Reprinted with permission from ref. [42]. Copyright 2014 Royal Society of Chemistry.

### 3.1.2. Carbon Nanotubes/Fibers (CNTs/CNFs)

With the ideal electrical conductivity, excellent tensile strength, and thermal conductivity, the CNTs/CNFs were widely applied as the air cathode material [43,44]. Furthermore, the CNTs/CNFs can be surface-modified with some chemical processes [45], boosting the potential application of many technologies. The CNTs were used as the cathode catalyst of the LABs, first reported by Reiner's group [46]. They found that CNTs revealed higher capacities than carbon black, graphite, and active carbon, especially the single-walled CNTs generated a capacity higher than 1600 mA h $g^{-1}$. A freestanding, hierarchically porous

carbon nanotube film was fabricated as a binder-free air electrode of the LABs (Figure 3a–c) [47]. With this air electrode, the $O_2$ can be transported rapidly through large open tunnels. The main product, $Li_2O_2$, can be formed and decomposed on the surface of the electrode with many substantial nanopores; the battery displayed a maximum specific capacity of 4683 mA h g$^{-1}$ and has excellent rate capability. Carbon nanotubes directly grown on nickel foam as cathodes of LABs exhibit a special capability of 12,690 mA h g$^{-1}$ at an ampere density of 200 mA g$^{-1}$ due to their intersecting wide channels, large hollow spaces, and seamless connections. Additionally, the long-cycle stability reached 110 cycles when the capacity was limited to around 2000 mA h g$^{-1}$ and the current density was 200 mA g$^{-1}$ (Figure 3d,e) [48]. Aligned multiwall nanotubes with controllable pore structures were woven into the CNT fibril as the air electrodes of the LABs revealed a capacity higher than 2500 mA h g$^{-1}$ during the first 20 cycles (Figure 3f–h) [49].

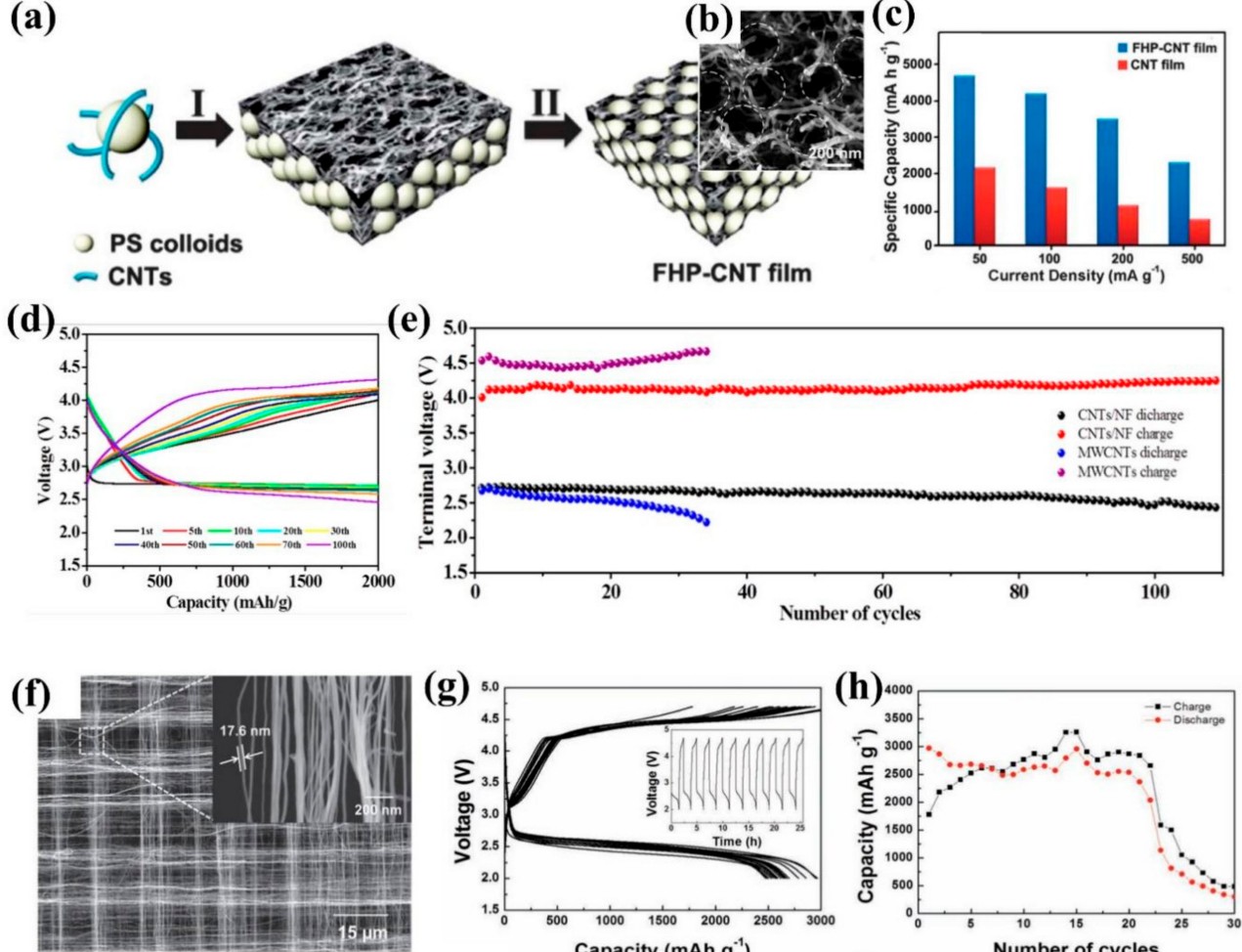

**Figure 3.** (**a**) Schematic diagram of multi-layer porous carbon nanotube film preparation process. (**b**) SEM images of interconnected large tunnels in CNT films. (**c**) Rate performance of pure carbon nanotube films and graded porous carbon nanotube films at different current densities Reprinted with permission from ref. [47]. Copyright 2013 Royal Society of Chemistry. (**d**) Discharge/charge profiles of CNT/NF in pure $O_2$ at 200 mA g$^{-1}$ with a cutoff capacity of 2000 mA h g$^{-1}$. (**e**) Cycling performance of CNT/NF and MWCNTs at a current density of 200 mA g$^{-1}$ with a cutoff capacity of 2000 mA h g$^{-1}$ Reprinted with permission from ref. [48]. Copyright 2016 American Chemical Society. (**f**) SEM images of CNT fibrils at high magnification. Discharge/charge curve. (**g**) and cycle capacity (**h**) of lithium oxygen battery using air electrode based on braided carbon nanotubes. Reprinted with permission from ref. [49]. Copyright 2013 John Wiley and Sons.

It is well known that when defects or impurities are introduced into the host material, the generation of active sites for electrocatalytic reactions can be promoted. The surface defects on carbon nanotube cathodes were created by ion bombardment in argon (Figure 4a–c) [50]. The SEM results indicated that on the surface of defective carbon nanotubes, $Li_2O_2$ has more active sites to nucleate and grow into tiny nanoparticles, which is beneficial to the recharging process of $Li_2O_2$ decomposition. The results revealed that the full discharge capacity increases and the overpotential decreases after the defects are created on the surface of the CNTs. Etched by the strong acid and alkali is another way to obtain the defective CNTs, such as carbon nanotubes with partially ruptured carbon nanotubes after etching with KOH at 800 °C for 120 min [51]. The etched CNTs showed that more active edge sites had been created to promote a high discharge capacity and better electrochemical performance. Shen et al. [52] decorated the carbon nanotube sponges with Pd nanoparticles as the air cathode, and its capacity and cycle performance were significantly improved compared with the bare carbon nanotube sponge. A perfluorinated alkyl chain was grafted to the CNTs, benefiting from the high $O_2$ affinity of perfluorinated moieties, and used as the air cathode. The results confirmed that the grafted structure could attract a very high $Li-O_2$ reaction and enhance the capacity and stability of the LABs [53]. Also, there are some other reports to decorate electrochemical properties of CNTs cathodes in $Li-O_2$ batteries [48,54–56].

Recently, carbon nanofibers (CNFs) were proven to be one of the ideal electrode materials to be used to assemble high-performance LABs. The CNFs can form flexible electrodes free standing, which suggests the most suitable use for the wearable electronic device. Combined with the electrospinning and the physical activation methods, freestanding activated carbon nanofibers (ACNF) were used for the cathode of the non-aqueous LABs [57]. In this paper, the number of sites for the $Li_2O_2$ nucleation increased after the $CO_2$ activation, and the size and micromorphology also changed. Compared to the non-activated CNF cathodes, the batteries with the ACNF electrode revealed a higher discharge capacity, lower overpotential, and better cycling stability. Well-aligned hollow carbon fibers synthesized on porous anodized aluminum oxide (AAO) substrates by a simple chemical vapor deposition (CVD) method can generate ultra-high gravimetric energy reaching 2500 Wh kg$^{-1}$ as LABs cathodes [58]. Inspired by the egg-laying process of the golden toad and its tightly packed eggs, Zhang's group [59] assembled a graded macroporous activated carbon fiber (MACF) electrode. Results displayed that the $RuO_2$ decorated MACF (R-MACF) and the MACF cathode reached a very high value of 13,290 mA h g$^{-1}$ and 11,150 mA h g$^{-1}$ at the ampere density of 1000 mA g$^{-1}$, respectively. This is higher than those for CNTs (around 6810 mA h g$^{-1}$), macroporous active carbon (MAC, about 5307 mA h g$^{-1}$), and active carbon fibers (ACFs, 1737 mA h g$^{-1}$). Most importantly, better cycling stability and distinguished rate capability were demonstrated with the MACF in addition (Figure 4d–g). Shui et al. [60] studied a $Li-O_2$ battery with the vertically aligned nitrogen-doped coral-like carbon nanofiber (VA-NCCF) array as the air electrode. This research shows that the fiber with a coral-like microstructure not only can provide more contacts between the electrode and the electrolyte but also offer more active-sites for $Li_2O_2$ nucleation. The ultimate result is the lowest overpotential of about 0.3 V, which was obtained at the ampere density of 100 mA g$^{-1}$ with a limited capacity of 1000 mA h g$^{-1}$ (Figure 4h–j). Besides, higher energy efficiency of around 90% and excellent cycle performance, which are far beyond outstanding than the undoped CNT and CNT powder cathode.

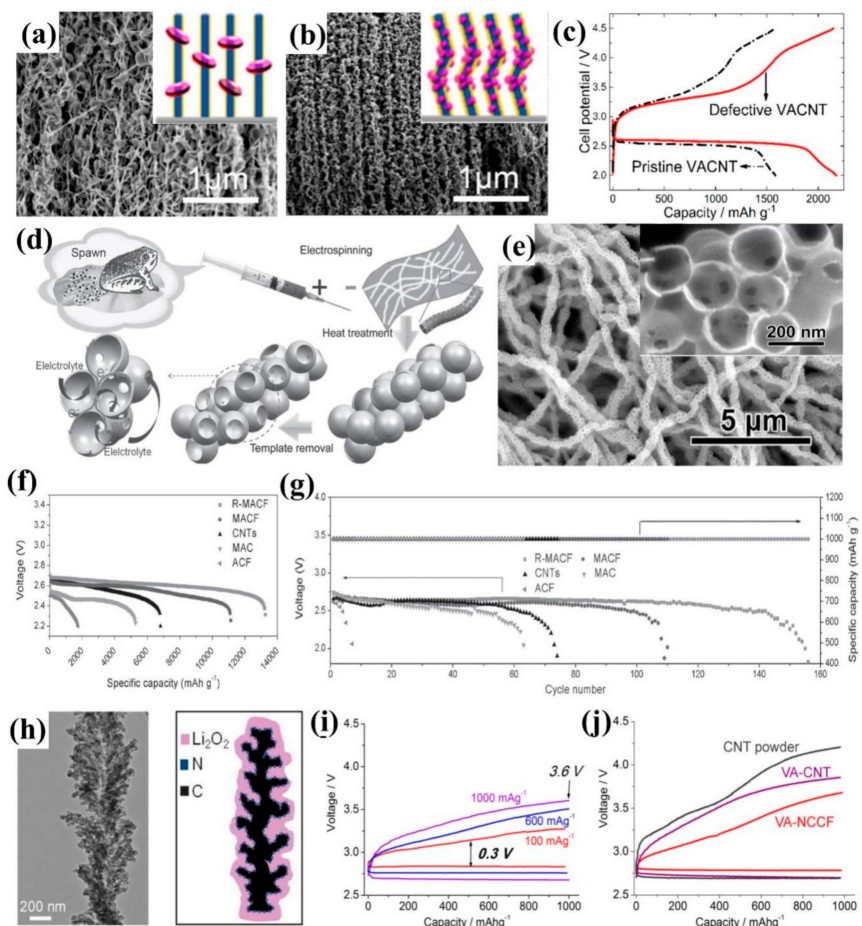

**Figure 4.** Original SEM image (**a**) and defects VACNT. (**b**) The cathode removed from the tetraethylene glycol dimethyl ether (TEGDME)-based battery after the first discharge process. (**c**) The first full cycle curve of N-methyl-N-propylpiperidinium bis(trifluoromethansulfonyl)imide (PP13TFSI) based battery with original and defective VACNT cathode was measured at an ampere density of 0.1 mA cm$^{-2}$ over the voltage range of 2.0–4.5 V. Reprinted with permission from ref. [50]. Copyright 2014 American Chemical Society. (**d**) Schematic diagram of preparation of the carbon fiber cathode. (**e**) SEM image of MACF. (**f**) At a current density of 1000, the rate capability of a lithium oxygen battery with five different cathodes and (**g**) the terminal discharge voltage of a lithium oxygen battery with five cathodes. Reprinted with permission from ref. [59]. Copyright 2016 John Wiley and Sons. (**h**) TEM images of a single VA-NCCF and sketches of Li$_2$O$_2$ deposited on coral like carbon fibers. (**i**) Rate characteristics of VA-NCCF electrode at current densities of 100, 600, and 1000 mA g$^{-1}$. (**j**) Comparison of VA-NCCF with undoped vertically aligned carbon nanotubes (VACNT) and CNT powders. Reprinted with permission from ref. [60]. Copyright 2014 American Chemical Society.

### 3.1.3. Graphene and Graphene-like Carbon Materials

Atomically thick graphene and graphene-like carbon materials have become very interesting topics in experimental and theoretical research on 2D nanomaterials, which can be used as highly efficient electrode materials for various energy conversion and storage devices. Theoretically, graphene is a single layer of sp$^2$ carbon atoms arranged hexagonally, is variously known as the oversized theoretical special surface area (~2630 m$^2$ g$^{-1}$), excellent intrinsic carrier mobility (200,000 cm$^2$ V$^{-1}$ s$^{-1}$), outstanding thermal conductivity (about 5000 W m$^{-1}$ K$^{-1}$), high optical transmittance (around 97.7%), and ranking mechanical strength [61–64]. Due to the above fantastic properties, graphene and graphene-like carbon materials were extensively applied as electrocatalysts of the cathode materials for the Li-O$_2$ battery [63–76]. Benefiting from the interlinked microporous channels of the unique bimodal porous structure, an exceptionally high discharge capacity of 15,000 mA h g$^{-1}$ was

confirmed in the LABs with the hierarchically porous graphene as the electrode [65]. Yoo and his coworker [66] designed a kind of metal-free graphene nanosheets as the electrode for the LABs. A low overpotential is about 0.56 V which can be attributed to $sp^3$ bonding and the edge/defect of graphene nanosheets. Moreover, the electrochemical properties of the LABs can be enhanced by adjusting the ratio of the $sp^2$ and $sp^3$ carbon atoms. Using graphene oxide (GO) as the carbon source and framework simultaneously, a freestanding and hierarchically porous carbon (FHPC) cathode was fabricated in nickel foam by Wang et al. [67]. A remarkable discharge capacity of 11,060 mA h $g^{-1}$ and 2020 mA h $g^{-1}$ could be detected with an ampere density of 280 mA $g^{-1}$ and 2.80 A $g^{-1}$, separately. Subsequently, various kinds of graphene or graphene-like carbon materials have been designed and fabricated for the application of the cathode catalysts of the LABs, such as porous graphene nanoarchitectures with various pore sizes, reduced GO (rGO), surface modified hierarchical graphene.

Highly flexible and porous free-standing graphene papers were fabricated and used as the cathodes of the LABs directly [68] with a higher discharge capacity of about 7000 mA h $g^{-1}$ (the capacity of the rGO paper and commercial carbon paper are 600 and 240 mA h $g^{-1}$, separately). Using the polystyrene spheres (PS) as the template, ultrathin and multi-wrinkled three-dimensional (3D) graphene sheets with an open and interconnected porous network were fabricated through a simple hydrothermal method and high-temperature annealing (Figure 5a,b) [69]. As the binder-free air electrode of the Li-$O_2$ batteries, an ultrahigh full discharge capacity of around 21,507 mA h $g^{-1}$ could also be detected at the ampere density of 200 mA $g^{-1}$, and an outstanding rate capability could be found. The pore diameter was investigated as an important parameter of the porous graphene impacting the electrochemical performance, and the optimized results were obtained in their report [70]. The holey graphene was used as the air cathodes for the non-aqueous Li-$O_2$ batteries, revealing a super-high areal capacity of about 40 mA h $cm^{-2}$ [71], and other closely linked research results hinted that the holey graphene was suitable as the cathode electrodes materials for the high-capacity and high-performance LABs. Through deoxygenation treatment, the mesoporous/macroporous structured hierarchical micron-sized graphene, as the cathode catalysts of the LABs, demonstrate an excellent full discharge capacity reaching 19,800 mA h $g^{-1}$, and it can work for more than 50 cycles at an ampere density of 1000 mA $g^{-1}$ with a cut-off capacity of 1000 mA h $g^{-1}$ [72].

The defects or impurities induced in the graphene-like carbon materials is an important part of the cathode catalysts for the LABs. For instance, nitrogen doping can change the electronic distribution of carbon materials, and more oxygen chemisorption sites can be detected on the surface of carbon materials [73]. Compared with the onion-like carbon (OLC), the nitrogen-doped onion-like carbon (N-OLC) revealed that the full-discharge capacity had been improved up to 12,181 mA h $g^{-1}$, and the overpotential had been reduced to about 0.88 V for the first cycle. The reduced overpotential during the discharge/charge processing led to round-trip efficiency increasing, and the value for the N-OLC and OLC cathode were 76% and 73%, separately. Moreover, the cell with the N-OLC electrode can loop for 194 cycles at the cut-off capacity of 1000 mA h $g^{-1}$ at an ampere density of 0.3 mA $cm^{-2}$ (Figure 5c) [73]. Furthermore, a hierarchical-structured 3D porous boron-doped reduction graphite oxide (B-rGO) was synthesized with a facile freeze-drying method [74]. The results revealed that B ions could promote the formation of cross-linked three-dimensional porous structure and the B-O functional group, which is conducive to the formation of more catalytic active centers and accelerate the diffusion of oxygen and the penetration of electrolyte. Finally, the electrochemical performance of the LABs was enhanced with the B-rGO electrode, especially for the fast charge with large current densities. In 2015, the nanocomposite graphene@g-$C_3N_4$ t composed of macroporous graphene and graphitic carbon nitride (g-$C_3N_4$) was first reported [75]. This was benefited by the macroporous graphene, which not only facilitated the electronic transport properties but also offered countless active sites for $Li_2O_2$ nucleation and decomposition. Electrochemical results show that the LABs have a lower overpotential of about 0.62 V and a full discharge capacity of

about 17,300 mA h g$^{-1}$. In addition, a charming cycle performance for 105 cycles with a cut-off capacity of 1000 mA h g$^{-1}$ at 0.4 mA cm$^{-2}$ would also be verified. From the above research, we know certain reliable truths regarding graphene-like carbon materials. For example, the edge effect and heteroatom doping can effectively improve the electron transmission capability, increase the catalytic activity sites, and enhance the electrocatalytic activity. In that case, what effect could be measured if the appealing traits of the heteroatom doping and the edge effects were integrated together? Under the guidance of this concept, porous nitrogen-doped holey graphene (N-HGr) was used as the cathode for rechargeable Li-O$_2$ batteries (Figure 5d–g) [37]. The holey graphene with many nanoscale holes can facilitate the doping of the nitrogen atoms that provide numerous valuable catalytic sites and more transmission channels for electrolyte/oxygen/electron transport with its rough edges. As expected by the author, the battery with the N-HGr cathode catalysts delivered a higher round-trip efficiency of around 85% and stable cycling performance of more than 100 cycles with a limited capacity of 800 mA h g$^{-1}$ at the ampere density of 40 mA g$^{-1}$, and a super higher full discharge capacity of 17,400 mA h g$^{-1}$ at 100 mA g$^{-1}$. Its electrochemical performance is better/more outstanding than most other carbonaceous air cathodes.

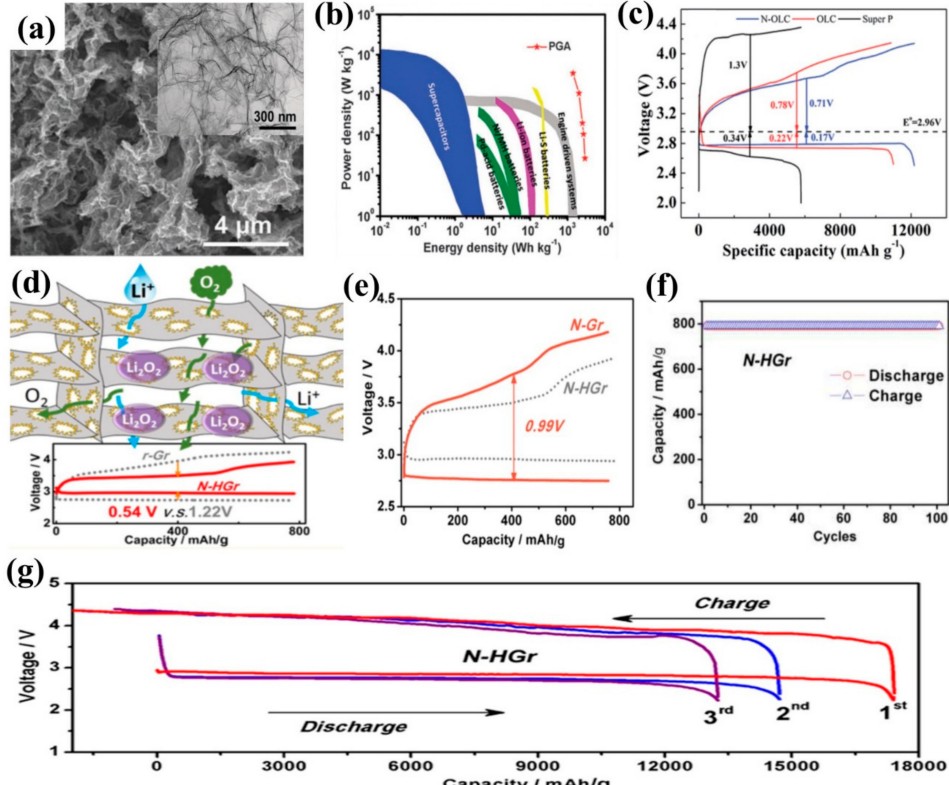

**Figure 5.** (**a**) SEM and TEM images of polystyrene sphere mediated ultra-thin graphene sheet assembled aerogel (PGA). (**b**) Ragone diagram of lithium oxygen battery with PGA as air electrode (based on the total mass of C + Li$_2$O$_2$). Reprinted with permission from ref. [69]. Copyright 2015 Royal Society of Chemistry. (**c**) First discharge/charge curve of lithium oxygen battery with NOLC, OLC and super P electrodes at 0.15 mA cm$^{-2}$. Reprinted with permission from ref. [73]. Copyright 2016 Royal Society of Chemistry. (**d**,**e**) Discharge/charge distribution of N-doped porous graphene (N-HGr), original graphene (r-Gr) and original porous graphene (r-HGr) at 40 mAg$^{-1}$. (**f**) Cycling stability of N-HGr with a limited capacity of 800 mA h g$^{-1}$ with a current density of 40 mA g$^{-1}$. (**g**) Three full discharge/charge cycles of N-HGr within a voltage window of 2.2–4.4 V under a current density of 100 mA g$^{-1}$. Reprinted with permission from ref. [37]. Copyright 2016 American Chemical Society.

In addition to nitrogen doping, sulfur-doped graphene was also studied. The results indicated that doping sulfur into graphene cathode could improve the cycle life and discharge stability in the laboratory. Note that this is different from nitrogen-doped graphene, which can mainly increase the discharge capacity of the batteries (Figure 6a–c) [76,77].

All of these nitrogen-doped graphene/graphene-like carbon materials revealed an obvious improvement in the electrochemical performance of the LABs, especially with the amazing enhancement of the ORR. However, their OER electrocatalytic properties are too difficult to be further applied in LABs. In terms of basic theory, a catalyst with the electron adsorption ability is an optimal catalyst material for OER and can enhance the electron transformation between the $Li_2O_2$ and the collecting base. In this respect, boron-doped graphite carbon shows more promise for the OER process of the LABs owing to the special acceptors' electronic structure of the boron. According to the first principle calculation results, the oxygen evolution barrier could be decreased by about 0.12 eV with the B-doped graphene, which greatly helps to improve the oxygen evolution rate and enhance the rate capability of OER [78]. Soon after, this theoretical feasibility was validated with the hierarchical structured 3D porous boron-doped reduction graphite oxide (B-rGO) materials that were designed and synthesized with a facile freeze-drying method (Figure 6d–f) [74]. As the cathode of the LABs, the 3D porous hierarchical structure can accelerate the diffusion/penetration of the oxygen or the electrolyte, and more edge or reaction sites obtained from the boron-oxygen functional groups could be detected on the graphene surface to activate the OER/ORR process. Otherwise, the π system of the carbon can be activated together with the stimulated boron ions that doped into the carbon lattices, which is beneficial for the fast-charging process of the batteries with large current densities. The final verification result shows that with the stronger interactions between the $Li_5O_6$ clusters and the B-rGO, the LABs with the B-rGO as a catalytic indicated a high discharge capacity, excellent rate capability, cycle stability, and overpotential reduction.

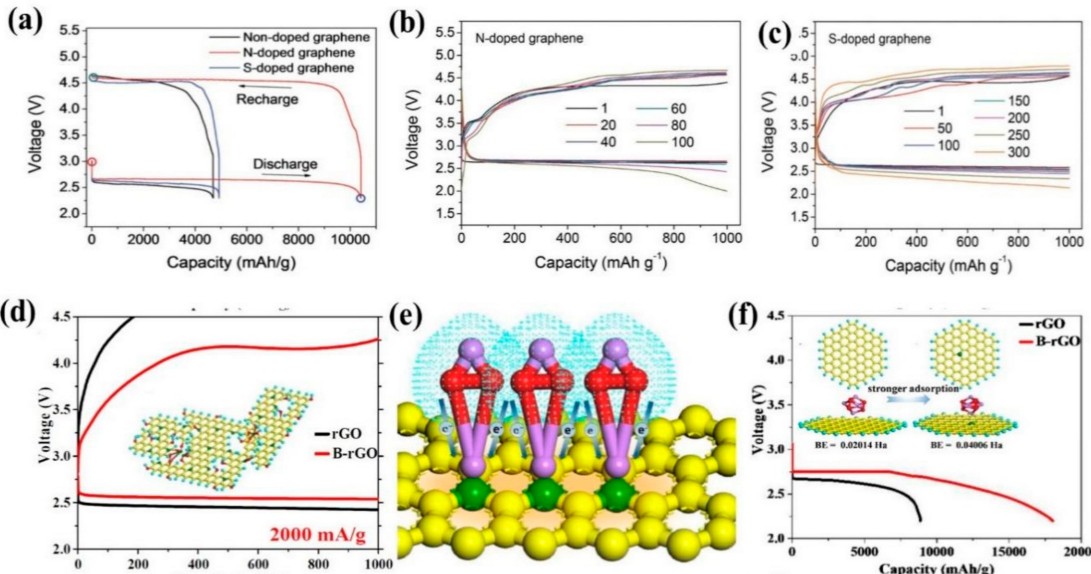

**Figure 6.** (**a**) Constant current discharge charge distribution of lithium oxygen battery with nanoporous graphene cathode at 200 mA g$^{-1}$ current density, (**b**,**c**) Charge–discharge cycling performances of the nanoporous N- and S-doped graphene for LABs with a cut-off capacity of 1000 mA h g$^{-1}$ at 300 mA g$^{-1}$. Reprinted with permission from ref. [76]. Copyright 2016 John Wiley and Sons. (**d**) First discharge charge distribution of lithium oxygen battery with B-rGO and rGO as cathode by reducing the capacity to 1000 mA h g$^{-1}$ 2000 mA g$^{-1}$. (**e**) Schematic diagram of obtaining electrons from $Li_2O_2$ using B-rGO as a substrate. (**f**) Discharge curves of rGO and B-rGO lithium oxygen Batteries as cathode to 2.2 V at current density of 500 mA g$^{-1}$. Reprinted with permission from ref. [74]. Copyright 2016 American Chemical Society.

### 3.2. Noble Metals

As noted above, although carbon materials have established themselves as the emerging candidate for the cathode catalysts for the LABs, there are still some shortcomings when the pure or doped carbon material were applied to boost the $Li_2O_2$ nucleation/decomposition process. A higher overpotential could be verified during the charge process that can be imputed to the sluggish kinetics of $Li_2O_2$ oxidization with the carbon catalyst, and the rate performance had been restricted. In order to cope with these short buckets, various noble metals, such as Pt, Pb, Au, Ag, Ru, and etc., and their oxides supported on carbon were used as cathode catalysts for the LABs. The uniformed noble metal (Ru and Pd) nanoparticles that were supported with the CNT fabrics were applied as the cathode catalysts for the LABs (Figure 7a–c) [79]. The Ru/Pd-CNT catalyst demonstrated remarkable electrochemical properties with a long and stable cycle life (more than 100 cycles), a charming coulombic efficiency, and a lower overpotential (0.53 and 0.83 V for Ru-CNT and Pd-CNT, respectively), which could equivalent or even better than some other best literature results. In Figure 7d–f, Ye et al. reported that the cycle performance was doubled when branched Pd nanodendrite-supported graphene nanosheets were used as cathode catalysts for LABs compared to graphene nanosheets [80]. Hollow carbon spheres that were decorated with the Pd nanocrystals (P-HSC) suggested a full-discharge capacity of around 12,254 mA h g$^{-1}$ at 500 mA g$^{-1}$ and a standing cycle-life more than 205 cycles at 300 mA g$^{-1}$ with a cut-off capacity of 1000 mA h g$^{-1}$ could be operated [81]. Comparing the value of around 125 cycles of the hollow spherical carbon (HSC) under the given conditions, the electrochemical performance of P-HSC was almost multiplied. $RuO_2$ as a shell covered on the surface of CNTs uniformly implied a high coulombic efficiency of about 79% with an overpotential of 0.72 V at an ampere of 100 mA g$^{-1}$, respectively. An excellent cycling performance over 100 cycles at a high ampere of 500 mA$^{-1}$ and a cut-off capacity of 300 mA h g$^{-1}$ could also be verified (Figure 7g–i) [28].

Lu's group synthesized 3D curved $Ag/Ag/NiO-Fe_2O_3/Ag$ hybrid nanofilms [82] as a carbon-free cathode of the LABs, and the specially designed structure reduced the overpotential and prolonged the capacity and stability of the battery. In addition, they reported that the trilayer-liked $Pd/MnO_x/Pd$ nanomembranes [83] displayed an expressive lowered charge overpotential around 0.2 V and with a remarkable coulombic efficiency of about 86% and an ultra-stable cycle of more than 269 cycles.

Varied bimetallic alloy nanoparticles of Pd supported with the nitrogen-doped reduced graphene oxide (N-rGO) were applied as the cathode catalyst for the LABs [84]. The results revealed that the added transition metal components enhanced the stability of the catalyst obviously, especially the catalyst with Fe (PdFe/N-rGO) proved to have the best stability. A long stability cycle of more than 400 cycles with a cut-off capacity of about 1000 mA h g$^{-1}$ at 400 mA g$^{-1}$ can be confirmed with the PdFe/N-rGO catalysts, which is much longer than the 80 cycles of the Pd/N-rGO cathode. The electrocatalytic properties of $Pt_3M$ bimetallic alloys (M = 3d–d transition metals) had been studied with first-principles calculations [85]. Here, the sluggish ORR and OER kinetics in LABs had been improved greatly, and the overpotential had been reduced obviously with the $Pt_3M$, especially the M belonging to the group 5 elements. The best electrochemical properties could be steamed from the abundant surface electrons of the group 5 elements in the $Pt_3M$ alloys. The designed catalyst was composed of $FeN_x$ dispersed on the network like hollow dodecahedral carbon and then modified with Ru nanoparticles ($FeN_x$-HDC@Ru). Because the uniformly dispersed $FeN_x$ fraction can promote ORR performance, while Ru nanoparticles can promote OER capability, charge transfer dynamics have been enhanced through the internal network like hollow structure, the low impedance $Li_2O_2$/catalyst contact interface can be obtained through the construction of Ru nanoparticles. These factors will lead to the effective acceleration of deposition and dissolution of $Li_2O_2$ during the discharge and charge process [86].

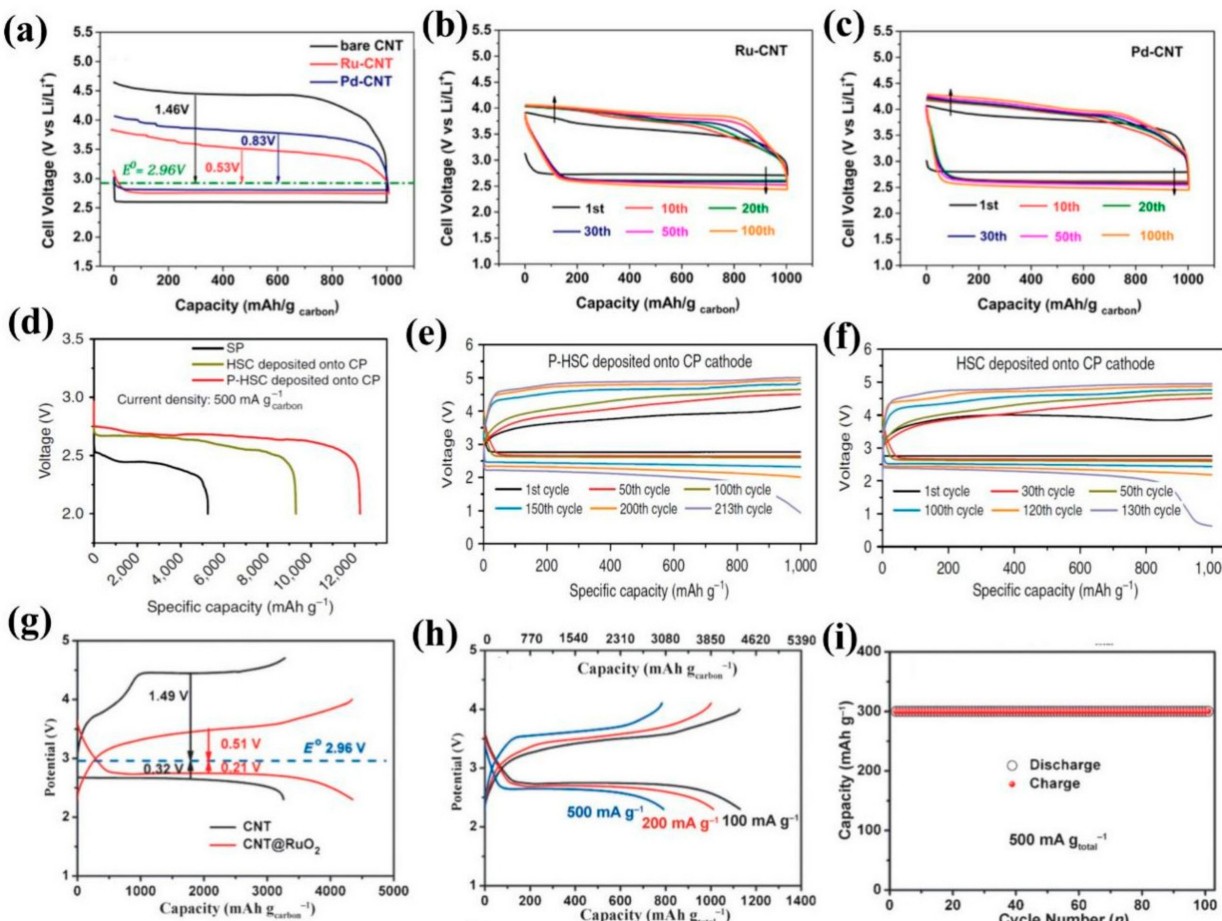

**Figure 7.** (**a**) Pristine CNT (black)s' first cycle, Ru-CNT (red), and Pd-CNT (blue) cathodes and multiple cycles of the catalyzed Li-O$_2$ cells based on (**b**) Ru-CNT and (c) Pd-CNT cathodes at 1000 mA h g carbon$^{-1}$. At 500 mA g carbon$^{-1}$. Reprinted with permission from ref. [79]. Copyright 2016 American Chemical Society. (**d**) Rate performance of three cathode lithium oxygen batteries at current density of 500 mA g$^{-1}$. (**e**,**f**) Discharge charge curve of lithium oxygen battery with P-HSC-HSC deposited on CP cathode under different cycles with 300 mA g$^{-1}$ and a cut-off capacity of 1000 mA h g$^{-1}$, separately. Reprinted with permission from ref. [80]. Copyright 2016 Royal Society of Chemistry. (**g**) Discharge/charge profiles of the Li-O$_2$ batteries with the pristine CNTs (2.3–4.7 V) and CNT@RuO$_2$ composite at a 385 mA g carbon$^{-1}$ (**h**). The first discharge/charge profiles of Li-O$_2$ batteries with CNT@RuO$_2$. (**i**) Cycling stability of the Li-O$_2$ battery at 500 mA g carbon$^{-1}$. Reprinted with permission from ref. [28]. Copyright 2016 John Wiley and Sons.

In addition to the above-mentioned research, there are many other reports, including various kinds of Ru and RuO$_2$-based materials [87–90] and other kinds of noble metals and their oxides [91–93], mainly as the nanocrystal-decorated porous graphene, carbon black supported noble nanocrystals, and noble nanocrystal-decorated vertical graphene nanosheets, etc. All are greatly helpful for ORR/OER performance during the discharge/charge process of the LABs, and it is difficult to include all aspects. However, although noble metals have good catalytic properties, the high material costs limit their applications.

### 3.3. Transition Metal Oxides

Due to their high prices, noble metals were immediately condemned in the eyes of candidates for cathode catalysts for the LABs. Hence, a low-cost and highly efficient cathode catalyst for ORR and OER has become one of the most urgent bugs to enhance the efficiency and lifetime of the LABs. Various kinds of non-noble metal materials catalysts,

such as the transition metal chalcogenides (mainly the transition metal oxides) and their nanocomposites, attracted a lot of interest from research and industry circles for their outstanding electrochemical catalytic performance and low price. Here, a brief discussion of the transition metal oxides and their nanocomposites with their advantages, such as acceptable excellent electrocatalytic performance, ease to repeat and make, low expense, will be unfolded.

### 3.3.1. Single Metal Oxides

Given the above-mentioned advantages, the transition metal and the transition metal chalcogenides have become promising candidates to replace noble metals for optimizing the sluggish ORR/OER during the electrochemical process. For electrochemically reversible processes of ORR and OER corresponding to dissociative or associative chemical reactions in LAB, in general, they undergo an electron transfer process between proton-coupled electrons and O* or OH* species [94]. Research in recent years has established that $Co_3O_4$, NiO, and $MnO_2$ are perceived as the most potential electroactive materials for LABs (Figure 8a–c) [95–102].

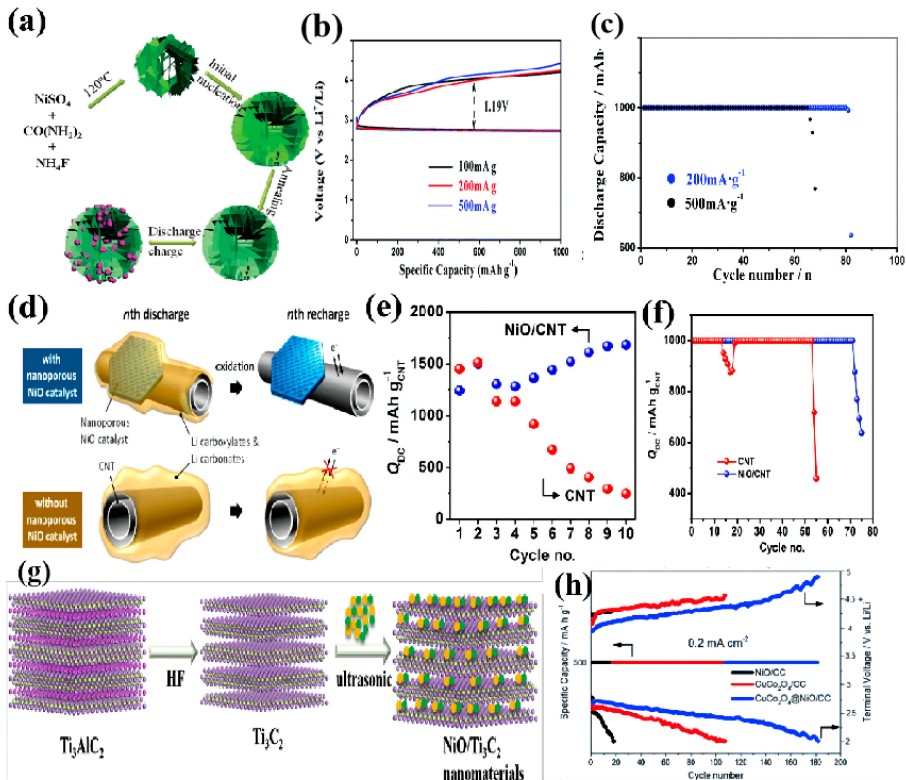

**Figure 8.** (**a**) Simple synthesis of flower like NiO and its application in rechargeable lithium oxygen batteries. (**b**,**c**) discharge/charge curve of NiO cathode lithium oxygen battery cycle performance under different current densities at 200 and 500 mA g$^{-1}$ with a fixed capacity of 1000 mA h g$^{-1}$. Reprinted with permission from ref. [98]. Copyright 2016 Royal Society of Chemistry. (**d**) Schematic diagram of carbonate/carboxylate decomposition of CNT and NiO/CNT catalyst electrodes. (**e**) Full discharge capacity retention of NiO/CNT and CNT electrodes with different cycle times at 0.1 mA cm$^{-2}$ current rate. (**f**) Discharge capacity retention of NiO/CNT (blue) and CNT (red) with a cut-off capacity of 1000 mA h g$^{-1}$ at 250 mA g$^{-1}$. Reprinted with permission from ref. [101]. Copyright 2015 American Chemical Society. (**g**) Schematic diagram describing the synthesis process of NiO/Ti$_3$C$_2$ nanomaterials. Reprinted with permission from ref. [104]. Copyright 2020 Elsevier. (**h**) cycle performance of NiO/CC, CuCo$_2$O$_4$/CC and CuCo$_2$O$_4$@NiO/CC. Reprinted with permission from ref. [103]. Copyright 2021 Royal Society of Chemistry.

NiO

Single-crystalline structured mesoporous NiO nanosheets as the $O_2$ electrodes of the LABs delivered a total discharge capacity of 1260 mA h $g^{-1}$ at 100 mA $g^{-1}$, and it can accomplish almost 40 cycles with a cut-off capacity of 500 mA h $g^{-1}$ [98]. Wang et al. found that a flower-like NiO exhibited a low discharge/charge potential gap of 1.19 V and long cycle life of 80 cycles with a cut-off capacity of 1000 mA h $g^{-1}$ at 200 mA $g^{-1}$ [96]. The two-dimensional (2D) hexagonal structured nanoporous NiO plates incorporated with CNT (NiO/CNT) electrode demonstrate a good cycle performance of more than 70 cycles with a cut-off capacity of 1000 mA h $g_{CNT}^{-1}$ at 250 mA $g_{CNT}^{-1}$ (Figure 8d–f) [99].

However, in past studies, the ORR performance of the NiO looks awkward and unstable in the LABs test, and there are few relevant reports even though it had superior OER performance. Recently, using different materials to modify NiO has been an effective method for improving the properties of the NiO cathode. Through the construction of a three-dimensional self-supporting $CuCo_2O_4$ nanowire@NiO nano chip core/shell array, the $CuCo_2O_4$@NiO/CC electrode can run 181 cycles with the capacity limit of 500 mA h $g^{-1}$ at 0.2 mA $cm^{-2}$ (Figure 8h) [103]. The $NiO/Ti_3C_2$ nanomaterial produced by Xing et al. showed the highest initial discharge capacity at 500 mA $g^{-1}$, up to 13,350 mA h $g^{-1}$, and the lowest OER and ORR overpotential when used as the cathode of the lithium–air battery. They believe that $Ti_3C_2$MXene ensures rapid electron transfer, and NiO nanoparticles provide enough catalytically active sites. The good dispersion of NiO nanocrystals on $Ti_3C_2$ greatly improves the concentration of active sites of $NiO/Ti_3C_2$ electrocatalyst to improve the catalytic activity of $NiO/Ti_3C_2$ nanomaterials (Figure 8g) [104].

Cobalt Oxides

Considering the eminent bifunctional catalytic performance of the cobalt oxides ($CoO_x$), especially the $Co_3O_4$ with spinel structure, they are identified as one of the most promising successors for the cathode catalysts of the LABs [101,105].

In recent years, various kinds of $Co_3O_4$ electrocatalysts used as the cathode of the LABs have been studied, including the $Co_3O_4$ cubes with different exposed planes [106], $Co_3O_4$ nanofibers [107], flower-like porous $Co_3O_4$ microspheres [108], $Co_3O_4$/carbon nanofiber composites [109], $Co_3O_4$/CNTs/CFP composite [110], $Co_3O_4$/Graphene composites [111], $Co_3O_4$/rGO [112–114], etc. (Figure 9a–c).

Some research results showed that the ORR and OER performance benefit from the bivalent and trivalent cobalt ions. Through these exposed $Co^{2+}$ and $Co^{3+}$ active sites, the $O_2/OH^-$ adsorption performance could be greatly enhanced. Therefore, the performance of bifunctional $Co_3O_4$ electrocatalysts can be optimized through complex design and ingenious arrangement of $Co^{2+}$ and $Co^{3+}$. Composited with the $Co_3O_4$ cube, which is exposed to the (0 0 1) plane, Gao and his group reported that the exposed (111) plane and the $Co_3O_4$ can reveal much higher specific capacity, better cycling performance, and good rate capacity [106]. Tomon et al. believed that the catalytic activity of $Co_3O_4$ depends on the proportion of $Co^{2+}$ in $Co_3O_4$. The site of $Co^{2+}$ can promote the formation of hydroxyl cobalt oxide, i.e., CoOOH, which is the key catalytic center of OER processes. They designed $Co_3O_4$ with different oxygen vacancy contents and obtained the best OER/ORR performance in the sample with the highest oxygen vacancy. The sample had the highest $Co^{2+}/Co^{3+}$ ratio (29.72%), and the catalyst provided the lowest charge/discharge potential gap (0.85 V) [115]. A unique nanoflower was assembled from $Co_3O_4$ nanosheets. This Co-flower has excellent OER properties and a unique porous 3D structure, conducive to the formation and decomposition of $Li_2O_2$. The Co-flower can greatly improve the long round-trip performance of LAB, as proved by 276 cycles at 0.5 A $g^{-1}$ charge/discharge current density and 248 cycles at 1 A $g^{-1}$. The long cycle life of LABs was attributed to the structure of $Co_3O_4$ nanoflowers, which provides a large space and rich electrocatalytic active sites for the formation and decomposition of $Li_2O_2$, especially at a high rate [116]. These results indicated that the reasonable shape-control or the exterior $Co^{3+}/Co^{2+}$ ratio control could enhance the electrocatalytic activity of the $Co_3O_4$ electrode materials.

Pt nanoparticles modified $Co_3O_4$ flakes arrays were synthesized and used as the cathode catalyst of the LABs; the result indicated that the recharged overpotential had been reduced from 4.0 V to 3.2 V that was contracted with the pristine $Co_3O_4$ arrays as catalysts [117]. Through laser-induced graphene (LIG) based processing, the $Co_3O_4$/LIG was produced and applied as an efficient cathode material for the LABs (Figure 9d–f). [118] An overpotential lower than 0.42 V can be detected in the first discharge/charge process at a current density of about 0.08 mA $cm^{-2}$ with a cut-off capacity of 430 mA h $g^{-1}$ and a full discharge capacity of about 240,000 mA h $g^{-1}$ could be detected for the first cycle. At a limited capacity of 430 mA h $g^{-1}$, 242 cycles were performed with these batteries, which indicated a long and stable battery cycle life. Therefore, varied $Co_3O_4$-based nanocomposites were studied as the catalytic cathode material for the LABs, such as the Ru-modified $Co_3O_4$ nanosheets, [119] Pd nanoparticles decorated $Co_3O_4$ nanosheets, [120] hetero-structured catalyst hybrid of the hexagonal $Co_3O_4$ coated with $MnO_2$ flakes, [121] and so on [122–124]. Mesoporous $CeO_2$/$Co_3O_4$ NWA was prepared on foamed nickel and was successfully used as a carbon-free and no adhesive electrode for LABs. The $CeO_2$/$Co_3O_4$ NWAS shows a high initial capacity of about 5539 mA h $g^{-1}$ at 50 mA $g^{-1}$ and more than 500 cycles at a capacity of 500 mA h $g^{-1}$ at 500 mA $g^{-1}$ [125]. $RuO_2$-$Co_3O_4$ nanohybrid with rich oxygen vacancies and a large specific surface area was developed as an electrocatalyst for LABs. The introduction of $RuO_2$ can increase the oxygen vacancy concentration of $Co_3O_4$. At the same time, due to the synergy between $Co_3O_4$ and $RuO_2$, it can accelerate the charge transfer and adjust the electronic structure to improve the activity of LABs. In addition, the unique hollow porous structure has a large specific surface area of about 104.5 $m^2$ $g^{-1}$, which can provide transmission channels for oxygen and electrolyte, provide space for discharge products, and expose more active sites (Figure 9g) [126]. $Co_3O_4$@3DOM $TiO_2$ electrode with hierarchical porous structure was prepared by in-situ encapsulation of $Co_3O_4$ in $TiO_2$ framework, which provided high capacity, good rate performance, and a durability of 7635.9 mA h $g^{-1}$ (288 cycles at 200 mA $g^{-1}$, limited specific capacity of 500 mA h $g^{-1}$). For $Co_3O_4$@3DOM $TiO_2$ electrodes, the thin-film discharge products can provide a large specific surface area, which is conducive to the rapid transmission of electrons at the catalyst interface. The interconnected macropores of the $TiO_2$ skeleton can provide sufficient transmission channels for electrolyte; $Co_3O_4$ NPs is embedded in the $TiO_2$ macroporous wall to prevent the agglomeration and flow away of $Co_3O_4$ NPs during long-term circulation; Mesopores on the wall of $TiO_2$ macropores can further promote the exposure of more active sites of $Co_3O_4$ (Figure 9h–j) [127]. However, beyond the $Co_3O_4$-$TiO_2$ electrodes showing the long cycle life, the porous urchin-like $Co_3O_4$ microsphere demonstrated a higher full discharge capacity of 9013.1 mA h $g^{-1}$ and good cycling stability for 200 cycles with a limited capacity of 500 mA h $g^{-1}$ at 400 mA $g^{-1}$ [128].

Except for the $Co_3O_4$, CoO-based materials also were implemented as a bifunctional cathode catalyst for rechargeable Li-$O_2$ batteries [129–132]. CoO mesoporous spheres were used as the cathode, indicating more than 300 discharge/charge loops with a limited capacity of 1000 mA h $g^{-1}$ at the current density of 0.04 mA $cm^{-2}$ [132]. Co-CoO nanoparticles embedded in the nitrogen-doped carbon nanorods (Co-CoO/N-CNR) were synthesized and utilized as the cathode demonstrated a full discharge capacity of 10,555 mA h $g^{-1}$ at 100 mA $g^{-1}$ [133]. Additionally, long cycle stability for over 86 loops was delivered with the Co-CoO/N-CNR at the limited capacity of 1000 mA h $g^{-1}$ with a current density of 100 mA $g^{-1}$. There are some other similar reports about the CoO-based materials, such as the CoO/Co nanocomposites on carbon nanotubes, [134] Co/CoO-Graphene-Carbonized melamine foam, [135] $Fe_3O_4$@CoO mesospheres [136]. However, compared to the $Co_3O_4$-based materials, a moderate gap could be detected when the CoO-based material was used as the electrocatalytic activity for LABs.

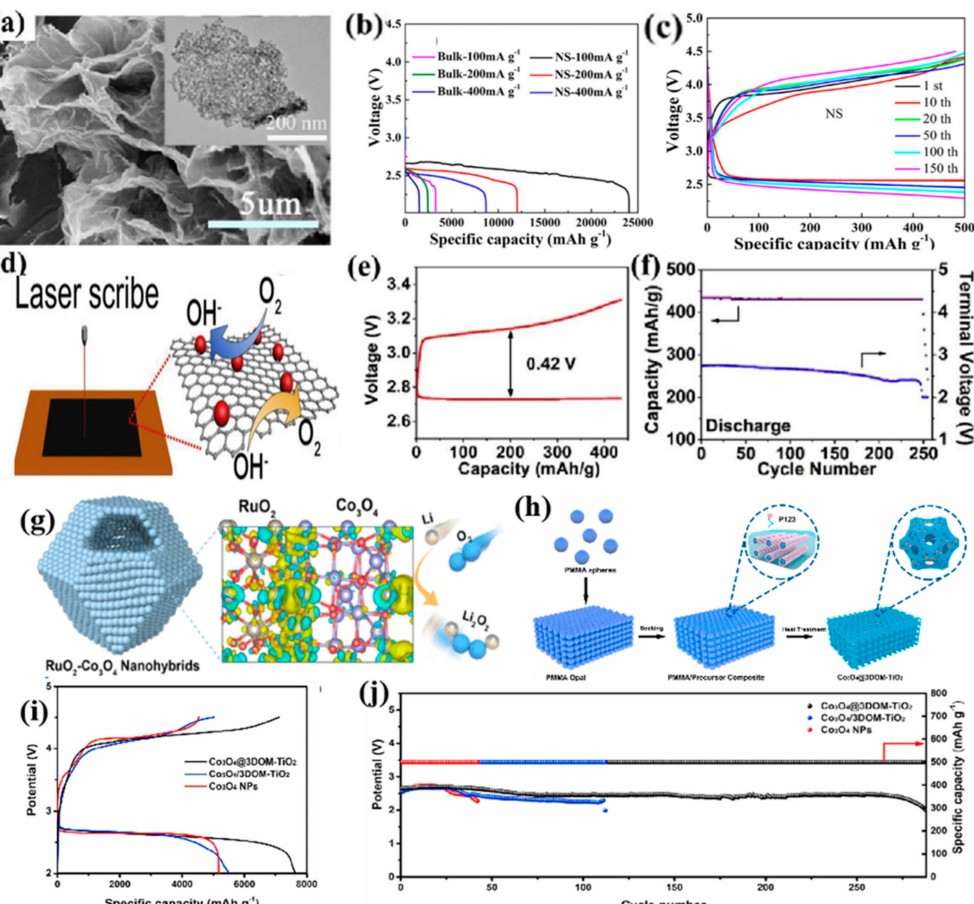

**Figure 9.** (**a**) SEM and TEM images of the $Co_3O_4$ nanosheets. (**b**) Specific capacities for the bulk and nanosheet samples at 100, 200, and 400 mA g$^{-1}$, respectively. (**c**) cycle performance of the $Co_3O_4$ nanosheets cathodes with a fixed capacity of 500 mA h g$^{-1}$ at of 400 mA g$^{-1}$. Reprinted with permission from ref. [113]. Copyright 2017 American Chemical Society. (**d**) Schematic illustration of the $Co_3O_4$/LIG. (**e**) first discharge and charge profile of the cyclic performance of the $Co_3O_4$/LIG electrode at a current density of 0.08 mA/cm$^2$ with cut-off capacity of 430 mA h g$^{-1}$. (**f**) discharge capacity and terminal voltage versus cycle number. Reprinted with permission from ref. [118]. Copyright 2018 Elsevier. (**g**) The working mechanism of $RuO_2$-$Co_3O_4$ nano hybrid. Reprinted with permission from ref. [126]. Copyright 2021 American Chemical Society. (**h**) Schematic diagram of the process of synthesizing $Co_3O_4$@3DOM $TiO_2$. (**i**) Initial deep discharge/charge curve at 100 mA g$^{-1}$. (**j**) Cyclic performance and terminal discharge voltage of $Co_3O_4$@3DOM $TiO_2$, $Co_3O_4$/3DOM $TiO_2$ and bare $Co_3O_4$ NPs at 200 mA g$^{-1}$, capacity cut-off to 500 mA h g$^{-1}$. Reprinted with permission from ref. [127]. Copyright 2021 Elsevier.

Manganese Oxides

On account of the fundamental structural unit of the $MnO_2$, various complex grids can be formed with the $[MnO_6]$ octahedron, and they have been differentiated into three categories: 1D structured $MnO_2$ mainly including α-, β-, γ-, and ε-$MnO_2$), 2D layer-liked δ-$MnO_2$, and 3D network structured spinel type λ-$MnO_2$ [102]. For the electrochemical performance of different phases of the $MnO_2$, the ranking shows that α-$MnO_2$ exhibits the best electrochemical performance, followed by β-$MnO_2$ and then γ-$MnO_2$ [137,138]. In this section, various structured $MnO_2$ electrocatalysts for Li-$O_2$ battery application will be discussed in detail.

The nanostructure of α-$MnO_2$ is more likely to be an effective catalyst for air cathodes because of its special internal large tunnel that can accept more discharge products. This means that there will be more space for the reversible reaction of the for-



mation/decomposition of the lithium oxides ($Li_xO_y$), and the diffusion could also be accelerated for the $O_2$ and $Li^+$. Bruce and his coworkers first presented that an initial discharge capacity as high as 3000 mA h $g^{-1}$ could be obtained with the $\alpha$-$MnO_2$ nanowires at 70 mA $g^{-1}$ [139]. Subsequently, $\alpha$-$MnO_2$ with various kinds of nanostructures, such as nanotubes, [140] nanorods, [141] hollow clews, [142] nanospheres, [143] and urchin-like morphology [144] have been produced and utilized as catalysts of the LABs. With a microwave-assisted hydrothermal reduction, $\alpha$-$MnO_2$ nanotubes have been synthesized [140]. When it was utilized as the cathode, the energy capacity and stability of the LABs were improved, compared with the $\alpha$-$MnO_2$ nanowires and the $\delta$-$MnO_2$ nanosheets. Adjusting the surface oxidation state is one of the important routes to improve the catalytic performance; hence, a lot of trivalence Mn ions have been reduced and exposed as the active site on the surface of the $\alpha$-$MnO_2$ nanowires [145]. When they were utilized as the cathode catalyst of the LABs, a full discharge capacity of up to 11,000 mA h $g^{-1}$ could be acquired at 200 mA $g^{-1}$ within the initial cycle, and the LABs show a superb rate capability also with a capacity of about 4500 mA h $g^{-1}$ at 5000 mA $g^{-1}$. Through a hydrothermal redox process, a 3D hollow $\delta$-$MnO_2$ framework was produced on the surface of the 3D N-doped carbon foam, and then the $\delta$-type was transformed into an $\alpha$-type. As the cathode catalyst of the LABs, the 3D $\alpha$-$MnO_2$ revealed a full discharge capacity of 8583 mA h $g^{-1}$ at 100 mA $g^{-1}$ and a charming rate capacity up to 6311 mA h $g^{-1}$ at 300 mA $g^{-1}$ could be also detected (Figure 10a–c) [146]. Along with a limited capacity of 1000 mA h $g^{-1}$, the battery demonstrated stable cycling up to 170 cycles at 200 mA $g^{-1}$. This result maybe is the most outstanding performance along with the reported based on the $MnO_2$-based electrocatalysts. Employed the Ni foam-supported ZnO arrays as templates, $\alpha$-$MnO_2$/carbon submicron tubes ($MnO_2$/CST) arrays were prepared and used as the cathode catalyst of the LABs. The composites exposed excellent cycle stability over 300 cycles at a limited capacity of 1000 mA h $g^{-1}$ at a current density of 800 mA $g^{-1}$ [147]. Pd nanoparticles and $\alpha$-$MnO_2$ nanowire loaded rGO sheet were prepared by a simple reduction method. Here, the synergistic effect of $\alpha$-$MnO_2$ nanowires and palladium nanoparticles was integrated by decorating on graphene to improve the recyclability and capacity, to obtain the high-efficiency performance of $Li$-$O_2$ battery. Prepared rGO/Pd/$\alpha$-$MnO_2$ mixed nanocomposite cathode shows that the initial discharge capacity is 7500 mA h $g^{-1}$ and can be stably cycled 50 times under the limited capacity of 800 mA h $g^{-1}$ [148]. Beyond that, the $\alpha$-$MnO_2$/$RuO_2$ [149], Au and Pd nanoparticles decorated urchin-like $\alpha$-$MnO_2$, [150] CP-$\alpha$-$MnO_2$-$Co_3O_4$ composite [151], and so on had been studied, but the performance keeps them in a mediocre state and still had the very big promotion space.

Although the narrow tunnels in the $\beta$-type $MnO_2$ are not beneficial to $Li^+$ diffusion, some theoretical results indicated that the oxygen vacancies play an important role in enhancing the electrocatalytic activity of $\beta$-$MnO_2$ [152]. However, there was no experimental evidence to certify the satisfied electrocatalyst performance with the $\beta$-$MnO_2$. Even when combined with a noble metal, such as the Au–Pd-loaded mesoporous $\beta$-$MnO_2$, [153] Ag nanoparticles decorated $\beta$-$MnO_2$ nanorods, [154] the electrocatalyst performance is still good. Even when combined with the $RuO_2$ nanoparticles, the $\beta$-$MnO_2$ nanorods indicated a reversible capacity of about 500 mA h $g^{-1}$ within 75 loops at 50 mA $g^{-1}$ [155].

Accompanied with many defects, vacancies, and nonstoichiometric Mn coordination, a special single or double chain which is a component of the single [$MnO_6$] octahedron, formed the unique structure of $\gamma$-$MnO_2$. As one of the promising oxygen catalysts, $\gamma$-$MnO_2$ has been widely measured as the cathodic catalyst of LABs, but the electrochemical performance was far from satisfactory, [156] even with noble metal decorations that contribute to a substantial performance improvement. Li et al. reported that the Ru/$MnO_2$/SP revealed a lower overpotential of about 0.32 V and excellent cycling stability, reaching 200 loops with a cut-off capacity of 1000 mA h $g^{-1}$ at 500 mA $g^{-1}$ (Figure 10d–f) [157]. The authors explained that the outstanding properties could be attributed to the synergistic effect of the hydrophobic ionic liquid and composite cathode.

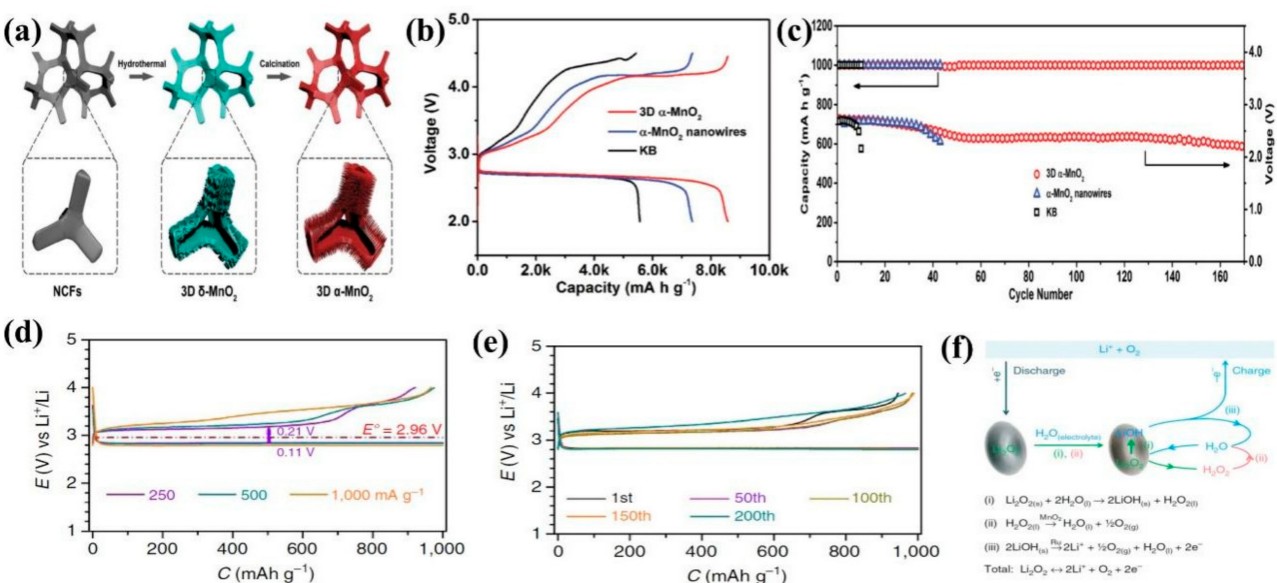

**Figure 10.** (**a**) Schematic diagram of 3D void α-MnO$_2$ frame fabrication. 3D void α-MnO$_2$ Discharge and charge curves (**b**,**c**) and cycle performance of MnO$_2$ nanowires and KB catalyst lithium oxygen battery at 100 mA g$^{-1}$ and 200 mA g$^{-1}$ with a limited capacity of 1000 mA h g$^{-1}$, respectively. Reprinted with permission from ref. [146]. Copyright 2019 John Wiley and Sons. (**d**) Discharge/charge profiles of the tenth run at varied Rate capability and (**e**) cycling performance of Li-O$_2$ battery with EMD/Ru/SP cathode. (**f**) Proposed reaction mechanism of EMD/Ru/SP cathode. Reprinted with permission from ref. [157]. Copyright 2015 Springer Nature.

The electrochemical performance had none of the sparkle of the cycle stability with the highly disordered structure of ε-MnO$_2$ and layered structured of δ-MnO$_2$. The sponge-like ε-MnO$_2$ directly expands on Ni foam, exhibits a full-discharge capacity of about 6300 mA h g$^{-1}$ at 500 mA g$^{-1}$), and enhances cyclability up to 120 loops [158]. Profit from the outstanding ORR activity of the γ-Fe$_2$O$_3$ and the excellent OER activity of the ε-MnO$_2$, Fe$_2$O$_3$@C@MnO$_2$ as the cathode catalyst of the LABs displayed an original discharge capacity up to 10,200 mA h g$^{-1}$ could be obtained. A narrow overpotential of about 0.67 V at 100 mA g$^{-1}$ and a stable prolonged cycle performance up to 260 cycles at 500 mA g$^{-1}$ could be confirmed separately, with a limited capacity of 1000 mA h g$^{-1}$ [159]. The 3D graphene (3D-G) and flower-like δ-MnO$_2$ (3D-G-MnO$_2$) air electrode exhibit superior specific energy of 9.7 KWh Kg$^{-1}$ and stable cycling life of about 132 times with low overpotentials at a 0.333 mA cm$^{-2}$ [160]. Homogeneous MnO$_2$ growth on a hierarchically porous carbon substrate (MnO$_2$/HPC) was employed as the cathode catalyst for the LABs [161]. Benefited from the unique intrinsic catalytic performance of the MnO$_2$/HPC, it revealed a reduced overpotential of about 0.68 V at 100 mA g$^{-1}$ and a stable loop-ability up to 300 cycles at 350 mA g$^{-1}$ with a cut-off capacity of 1000 mA h g$^{-1}$. Moreover, a charming rate capacity up to 2260 mA h g$^{-1}$ at 5 A g$^{-1}$ could be attained as well.

The crystal structure of the MnO$_2$ has a decisive effect on the catalytic activity. Benefiting from the large open tunnels and the exposed characteristic crystal plane, α-MnO$_2$ shows the best electrocatalytic properties. For these advantages, they did not show up on the β-, γ-, and ε-phase MnO$_2$. However, the special layer structure can remedy the defect of fewer tunnels and exhibit outstanding electrocatalytic performance. Furthermore, the morphology, size, pore structure, and compositions of MnO$_2$-based nanocomposite materials also critically impact the electrochemical performances of the cathode. Therefore, with the flexible adjustment of the micro/nanostructure, morphology, and composition, the optimized MnO$_2$-based cathodes could increase the performance of the LABs even further.

Besides the $MnO_2$-based cathode MnO [162], $Mn_3O_4$ [163], and $Mn_2O_3$ [164] are also being investigated as the cathode catalyst of the LABs. However, relative to the $MnO_2$-based electrode, the performance still needs to be further improved.

### 3.3.2. Polymetallic Oxides

With the above exposition, more active sites can prove better electrocatalyst performance, and other doped/mixed ions, especially metal ions, are an effective approach to get more active sites. Moreover, the mixed metal ions can enhance the electrochemical performance with the synergistic effect between the different metal ions. Hence, polymetallic oxides with superior inherent activity have been regarded as one of the high-efficiency catalysts for electrochemistry. Furthermore, the electrocatalyst activity of the polymetallic oxides could be modulated and optimized easily via the adjustment of the compositions, the crystalline structure, vacancy, electronic states, and the valence of the different metal elements. In this part, some polymetallic oxide high-efficiency bifunctional catalysts with spinel and perovskite structure and their excellent performance that were applied as the electrocatalysts of the LABs will be discussed.

Spinel Oxides

$A_xB_{3-x}O_4$ is the typical composition of the spinel polymetallic oxide, which has eight tetragonal subunits forming a cubic unit cell accompanied by many disordered sites caused by cation arrangement. This special structure with the tiny structural changes, such as the x in this polymetallic oxide has a small change and so on, has a great influence on the performance of the ORR and OER [101].

It is well known that the Co/Mn-based spinel oxide demonstrated electrocatalytic activity and stability in the basic electrolyte. They were considered the most promising bifunctional oxygen catalysts for the LABs. The $MnCo_2O_4$ nanowires anchored on the rGO nanosheets were applied as the cathode air electrode of the LABs, [165] and the results showed that the initial full discharge capacity was as high as 11,092.1 mA h $g^{-1}$ and the Coulombic efficiency was as high as 99.6%.

$Co_3O_4$, a special spinel structure, had been summarized and exhibited outstanding performance as the cathode catalysts of the LABs. However, it is well known that $Co^{2+}$ is a toxic substance that harms environment. Therefore, some metal ions with environmentally friendly characteristics are used to replace $Co^{2+}$ and improve the electron transmission performance [166–168]. A reduced overpotential and improved capacity of the spinel $MCo_2O_4$ were demonstrated. In particular, the $FeCo_2O_4$ electrode with the highest $Co^{3+}$ ratio which benefited from the $Fe^{2+}$ in the tetrahedral site (which releases electrons to reduce oxygen and form the $Fe^{3+}$ with half electron filled facilely) displayed the minimum overpotential, highest capacity, and best cycling performance.

Wu et al. designed and synthesized the three-dimensional $MnCo_2O_4$ nanowire with a high efficiency hierarchical porous structure via the hydrothermal method (Figure 11a–d) [169]. When it was applied as a carbon-free and binder-free cathode for the LABs, an original full discharge capacity was higher up to 12,919 mA h $g^{-1}$ at 0.1 mA $cm^{-2}$ could be detected. Meanwhile, the first full discharge capacity as 10,146, 7112, and 4771 mA h $g^{-1}$ could be tested when they were fitted at the higher rates of 0.2, 0.5, and 1.0 mA $cm^{-2}$, respectively, which indicated a superior rate capacity of the $MnCo_2O_4$ nanowire catalysts. When acting as the cathode of the LABs, an overpotential of about 0.54 V could be measured with a cut-off capacity of 500 mA h $g^{-1}$ at 0.1 mA $cm^{-2}$, and the cell can operate for over 300 cycles in this condition. Even with a constrained specific capacity of 1000 mA h $g^{-1}$, 144 stable charge and discharge cycles could be detected cycles at 0.1 mA $cm^{-2}$. The author considers that the special hierarchical interconnected mesoporous/macroporous weblike structure of the hybrid $MnCo_2O_4$/Ni foam cathode enhances the capability of electron transmission directly, which is the main reason for its excellent electrochemical performance.

Combined with the traditional hydrothermal method and subsequent calcination process, hierarchical $CuCo_2O_4$ flowers (CCF) were synthesized and assembled with porous

$CuCo_2O_4$ nanosheets [170]. When they were applied as the cathode of the LABs, a reduced overpotential of about 0.90 V could be measured with a limiting capacity of 1000 mA h $g^{-1}$ at 100 mA $g^{-1}$, and the result shows stable charge/discharge profiles for 120 cycles at the current density of 100 mA $g^{-1}$ could also be demonstrated (Figure 11d–g).

Due to the excellent catalytic activity and the mesoporous structure, the $NiCo_2O_4$ nanoflakes were used as the cathode catalyst and presented a reduced overpotential and enhanced cyclability [171]. Palani et al. prepared three-dimensional (3D) porous $NiCo_2O_4$ dodecahedral nanosheets (NCO) from an organic metal framework template (ZIF-67) and combined them with two-dimensional (2D) multilayer graphene nanosheets (GNs). The excellent performance of NCO@GNS-based cathode can be attributed to the existence of $CO^{3+}/CO^{2+}$ and $Ni^{3+}/Ni^{2+}$ redox pairs providing high bifunctional activity for ORR and OER processes, while high surface area provides many three-phase interfaces and active sites. Additionally, porous and dodecahedral nanosheets have excellent crystallinity. Cubic spinel NCO is attached to the GNs mixing platform to accelerate electron transport. Its unique 3D/2D graded mesoporous/microporous structure not only promotes $O_2$ diffusion, $Li^+$ ion transport and electrolyte penetration but also provides space for the deposition of discharge products (Figure 11h) [172]. Furthermore, Fe-doped binary $NiCo_2O_4$ was prepared by hydrothermal synthesis and annealing (NiCoFeO@NF). Fe with different 3D electron numbers can adjust the electronic structures of Ni and Co. Specifically, the 3D 5 electronic configuration of $Fe^{3+}$ is suitable for the coordination of octahedral or tetrahedral sites, while the 3D 6–8 electronic configuration of Co and Ni ions is conducive to the energy stabilization of octahedral coordination crystal field. Based on NiCoFeO@NF, the battery has a high discharge-specific capacity of 16,727 mA h $g^{-1}$ and excellent long-term cycle stability of more than 790 h at a current density of 500 mA h $g^{-1}$ (Figure 11i–k) [173].

With the two-step method, Cao and other team members designed and constructed the $MnCo_2O_4$ and P-doped hierarchical porous carbon hybrid named MCO/P-HPC [174]. The MCO/P-HPC hybrid acted as the cathode catalyst and presented a high full discharge capacity of up to 13,150 mA h $g^{-1}$ at 200 mA $g^{-1}$, and an outstanding rate capability of about 7028 mA h $g^{-1}$ at 1000 mA $g^{-1}$ was also revealed. With a limited capacity of 1000 mA h $g^{-1}$, the battery demonstrated a cycle stability of up to 200 loops at 200 mA $g^{-1}$. These results indicated that the chemical and electrical coupling between the $MnCo_2O_4$ and P-doped carbon played a critical role in enhancing the electrochemical catalyst performance. Moreover, the hierarchical macro-mesoporous structure of the electrode contributed to the improvement of the performance by adding more active sites and increasing the contact area between the electrolyte and active substance. In addition, it provides more places for $Li_2O_2$ nucleation and growth. The spinel stretched $LiMn_2O_4$ nanoparticles were decorated on the nitrogen-doped reduced graphene oxide (LMO@N-rGO) via a facile hydrothermal process [175]. The LABs batteries that were assembled with the LMO@N-rGO cathode displayed a lower charge plateau of about 0.21 V and a higher capacity of up to 7455 mA h $g^{-1}$ at 250 mA $g^{-1}$. When it operated with a cut-off capacity of 1000 mA h $g^{-1}$, the battery could work for 120 stability cycles at a current density of 375 mA $g^{-1}$. This outstanding performance can be attributed to the highly uniform dispersion of LMO nanoparticles on the surface of the N-rGO. Porous $MnCo_2O_4$/graphene (MCO/G) hybrids were prepared via a sonochemical method and applied as a cathode material in non-aqueous $Li-O_2$ batteries. The results indicated that a reduced discharge/charge overpotential of about 0.8 V could be detected and the cell could operate for 250 stability cycles with a fixed capacity of 1000 mA h $g^{-1}$ at 800 mA $g^{-1}$ [176]. The author believes that the outstanding performance of the batteries could be attributed to the coefficient of the mesoporous $MnCo_2O_4$ nanospheres and graphene sheets. The ORR/OER activity was derived by the mesoporous $MnCo_2O_4$ nanospheres, and the electrons/ions could be transferred with the graphene sheets. This synergistic effect led to an acceleration formation and the decomposition of the lithium peroxide. Waxberry-like hierarchical $NiCo_2O_4$ nanorods decorated on the surface of the carbon microspheres (NCO@CMs) exhibited a full initial discharge capacity higher, up to 6489.5 mA h $g^{-1}$, and an outstanding coulombic efficiency as high as 93.7%

at 200 mA g$^{-1}$ [177]. Ultrathin porous NiCo$_2$O$_4$ nanosheets with oxygen vacancies had been worked as the cathode catalysts of the LABs. And a superior full discharge capacity up to 16,400 mA h g$^{-1}$ at 200 mA g$^{-1}$ could be obtained in the initial cycle [178]. With the restricted capacity of 1000 mA h g$^{-1}$, the battery-operated stability for 150 cycles, and a special solution mechanism of the electrochemical double layer at the interface of cathode and electrolytes illustrated the charming electrochemical performance. Further, our group synthesized a microsphere containing NiCo$_2$O$_4$@CeO$_2$ composite. The unique microstructure catalyst can offer enough electrocatalytic surface to promote the barrier-free transport of oxygen and Li$^+$. Moreover, the feature microsphere and porous structure could effectively accelerate the penetration of electrolytes and the reversible deposition and dissolution of Li$_2$O$_2$. At the same time, the introduction of CeO$_2$ could enhance the oxygen vacancy and optimize the electronic structure of NiCo$_2$O$_4$ so as to enhance the electron transport of the whole electrode. This catalytic cathode material could effectively reduce the over potential to only 1.07 V and has a significant cycle stability of 400 cycles at 500 mA g$^{-1}$ [179].

Beyond that, there are many other spinel structured materials, such as CuCo$_2$O$_4$ nanotubes [180], MnCo$_2$O$_4$ nanotubes [181], MnCo$_2$O$_4$/MoO$_2$ nanocomposites [182], etc., which revealed the excellent catalyst performance for the LABs, which will not be discussed in detail here.

Spinel structure with the flexible crystal structure can easily be doped to change the catalytic performance, but the relationship between structure and catalytic activity is not precisely clear. Tracing and follow-up work on the mechanism of the catalytic activity with the development of the structural chemistry and the controllable preparation methods still is the key point of the applied research for the LABs.

Perovskite Oxides

Attributing to the outstanding OER performance of the perovskite oxides, it had aroused a vast concern all over the world, and the ORR performance of this kind of material has been developed more thoroughly in recent years. However, there is some room for improvement to match the superior OER activity. As everyone knows, the ABO$_3$ is the typical structural formula of the perovskite oxide composites, where A denotes the rare earth or alkali element, and B is the transition metal that plays a decisive role in the properties of the perovskite materials. For instance, the B sites usually act as the active center when they are used as the electrocatalyst [183]. One principle is that focus on the σ*-orbital (e$_g$) occupation and the extent of the B-site transition-metal–oxygen covalence has been demonstrated by the charming ORR/OER performance of the perovskite oxides [184,185]. Based on this theory, a series of perovskite oxide were prepared and applied as the cathode catalysts of the LABs.

Due to the good electrocatalytic performance of CaMnO$_3$, porous structured CaMnO$_3$ revealed a reduced overpotential of about 0.62 V at 50 mA g$^{-1}$ when they were applied as the cathode catalysts of the LABs [186]. Due to the synergistic effect of the porosity and higher catalytic activity of LaFeO$_3$, a stable cycle performance of up to 124 cycles could be detected with the of the 3D ordered macroporous LaFeO$_3$ cathode (3DOM-LFO) [187]. Porous LaNiO$_3$ nanocubes were used as the air electrode of the LABs and revealed a stable cycle of up to 75 cycles with a controlled capacity of 500 mAh g$^{-1}$ at 0.08 mA cm$^{-2}$ [188].

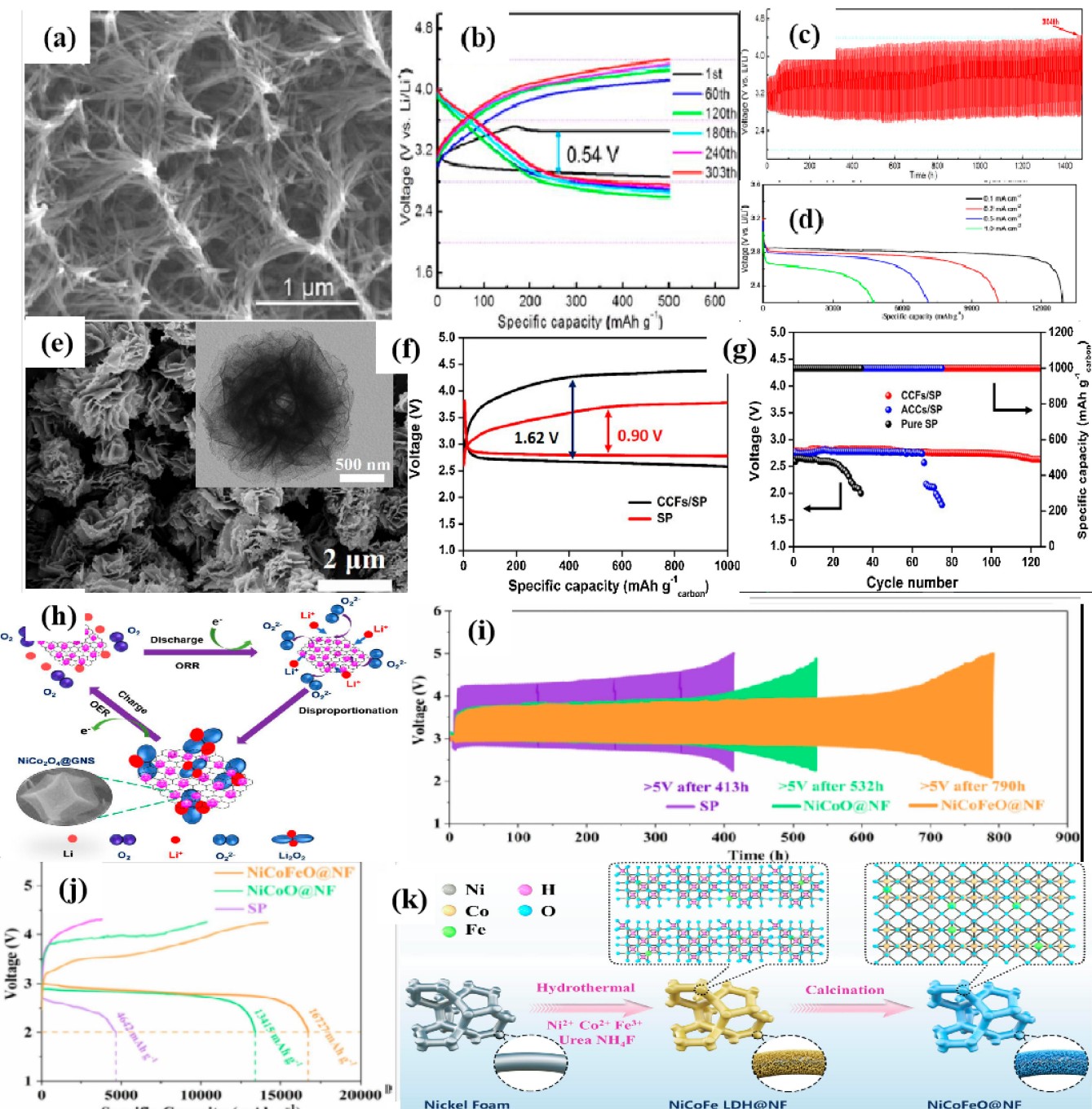

**Figure 11.** (**a**) Surface morphology and selected area electron diffraction (SAED) spectra of MnCo$_2$O$_4$ samples. (**b**) Constant current discharge/charging curve of different cycles and (**c**) discharge/charge voltage profiles with a fixed capacity of 500 mA h g$^{-1}$ at 0.1 mA cm$^{-2}$, (**d**) rate capability of the Li-O$_2$ cell with the MnCo$_2$O$_4$ electrode. Reprinted with permission from ref. [168]. Copyright 2017 American Chemical Society. (**e**) SEM & TEM images of the as-obtained CuCo$_2$O$_4$, (**f**) Discharge/charge curves and (**g**) voltage on the terminal of discharge versus cycle number of LABs with CCFs/SP and pure Super P electrodes at a current density of 100 mA g$^{-1}$. Reprinted with permission from ref. [169]. Copyright 2016 Elsevier. (**h**) Schematic diagram of NCO@GNS catalysts' formation/decomposition of Li$_2$O$_2$ by-products during discharge/charge cycle. Reprinted with permission from ref. [172]. Copyright 2021 Elsevier. (**i**) Cycle performance of NiCoFeO@NF base LABs at 500 mA g$^{-1}$, capacity 1000 mA h g$^{-1}$. (**j**) Constant current discharge curve of battery. (**k**) Preparation of NiCoFeO@NF nanowire arrays. Reprinted with permission from ref. [173]. Copyright 2022 Elsevier.

Recently, research results indicated that with the filling of the $e_g$-orbital antibonding states increasing, the covalence of the B-O band could be increased at the same time, leading to the enhancement of catalytic activity for the ORR/OER [184]. Furthermore, the catalytic activity can be improved with the bounded oxygenated species on the surface of the B-sites [189]. Therefore, various kinds of co-doped double perovskite oxide have been studied. The mesoporous structured LaSrMnO composited with the graphene foams revealed a ten full discharge/charge process with the discharge capacity kept at $2500-4500$ mAh g$^{-1}$ at 500 mA g$^{-1}$, and high-capacity retention of up to 55% can be detected [190]. When they were tested with a cut-off capacity of 500 mAh·g$^{-1}$, the cell could operate stably for 50 cycles, and the coulombic efficiency was nearly 100%. Perovskite structured $LaCo_{0.8}Fe_{0.2}O_3$ nanowires combined with the reduced graphene oxide sheets ((LCFO@rGO) were used as the cathode of the LABs [191]. The results explored the notion that a superior full discharge capacity of up to 7088.2 mA h g$^{-1}$ could be detected at the first cycle. The battery could be operated stably for over 56 cycles with a controlled capacity of 500 mA h g$^{-1}$ at 200 mA g$^{-1}$, and a low overpotential of about 0.98 V with a large round-trip efficiency could also be measured. The excellent performances of the nanocomposites could be attributed to the synergy effects between the LCFO and rGO as well as the inherent catalytic activity of the perovskite and the outstanding electron transmission capability of the rGO. Due to its unique porous nanotube morphology, LCFN was prepared through electrospinning and heat treatment as a catalyst for LABs. LCFN nanotubes showed excellent catalytic activity for OER and ORR, and there was no accumulation of $Li_2O_2$ during the discharge/charge cycle. Because the special structure and considerable catalytic activity of LCFN, the corresponding LABs can provide a discharge capacity of up to 13,019 mA h g$^{-1}$ and obtain a long cycle performance of 139 cycles at 400 mA g$^{-1}$ [192].

The segregation acted as a critical role in affecting the electrocatalytic performance of the perovskite oxides. Hence, segregation of perovskite was reported by Cong et al. to improve the efficiency of the cathode catalyst for the LABs. With the designed A-site cation deficiency, the $LaFeO_{3-x}$-supported $\alpha$-$Fe_2O_3$ was obtained via the annealing of the parent $La_{0.85}FeO_{3-\delta}$. As the properties of the oxygen adsorption were improved with the special structure of the $\alpha$-$Fe_2O_3$-$LaFeO_{3-x}$, a higher discharge specific capacity, reduced overpotential, and the rete capability were enhanced [193]. The cycle stability had especially been prolonged to 108 cycles with a limited capacity of 500 mA h g$^{-1}$ at 100 mA g$^{-1}$, which is almost four times that of the $LaFeO_3$ electrode.

Recently, the perovskite oxides composited with $RuO_2$ were investigated as cathode catalysts for LABs. For instance, the $RuO_2$ nanosheets decorated with the porous $La_{0.6}Sr_{0.4}Co_{0.8}Mn_{0.2}O_3$ nanofibers [194] exhibited a specific discharge capacity higher up to 12,741.7 mA h g$^-$, and a steady cycle of 100 cycles can be detected at 50 mA g$^{-1}$ with a controlled capacity of 500 mA h g$^{-1}$ also. $RuO_2$ nanoparticles modified Perovskite $La_{0.6}Sr_{0.4}Co_{0.2}Fe_{0.8}O_3$ nanofibers ($RuO_2$@LSCF-NFs) [195] were applied as the cathode catalyst for LABs and implied an outstanding cycling life for up to 120 loops with a cut-off capacity of 1000 mAhg$^{-1}$ at 100 mAg$^{-1}$ (Figure 12a–c). One nanocomposite composed of 0D $RuO_2$ NPs, 1D perovskite $LaMnO_3$ nanofibers (NFs), and 2D graphene nanosheets displayed an outstanding cycle life of over 320 cycles without voltage degradation. The outstanding performance could be attributed the advantages of each individual component [196,197].

The Ni-doped $La_{0.8}Sr_{0.2}Mn_{1-x}Ni_xO_3$ nanoparticles with abundant oxygen vacancies [198,199] operated as the cathode catalysts of the LABs, exhibiting a capacity of about 5364 mA h$_{gcarbon}$$^{-1}$, a reduced overpotential, and a lifespan prolonged to 79 cycles, which is longer than the $La_{0.8}Sr_{0.2}MnO_3$ catalyst. Du et al. prepared porous perovskite nanoparticles LSCFO by Co-doped with Sr and Fe cations in $LaCoO_3$ ($La_{0.8}Sr_{0.2}Co_{0.8}Fe_{0.2}O_{3-\sigma}$). Due to its excellent dual function electrocatalytic activity, the LABs based on LSCFO provide a high discharge specific capacity of up to 26,833 mA h g$^{-1}$ at 50 mA g$^{-1}$, an ultra-low charge/discharge overpotential of 0.32 V, and excellent long-term cycle stability of more than 200 cycles, with a limited capacity of 1000 mA h g$^{-1}$ at 300 mA g$^{-1}$. The significant

electrochemical performance can be partly attributed to the large specific surface area and unique mesoporous structure, which is conducive to exposing rich active sites and establishing effective mass transfer channels (Figure 12d,e) [200].

Recently, $CsPbBr_3$ nanocrystals were used as high-efficiency cathode catalysts for high-performance LABs for the first time. Among the reported perovskite catalysts, the LABs based on $CsPbBr_3$ have the lowest charging overpotential (0.5 V) and can maintain 400 stable cycles when the capacity is fixed at 1000 mA h $g^{-1}$, and the current density is 0.5 A $g^{-1}$. The weak adsorption strength between $Li_2O_2$ and $CsPbBr_3$ is the main reason for the low charging overpotential of $CsPbBr_3$-based LABs (Figure 12f,g) [201].

In short, due to the charming ORR/OER activity, exploring the perovskite structure oxides as the electrocatalysts for Li-$O_2$ battery is quite appealing. In addition, the surface properties of perovskite structure oxides are a key to understanding the origins of their electrocatalytic mechanism and working out the catalyst's materials with high efficiency via moderate and controllable synthetic methods.

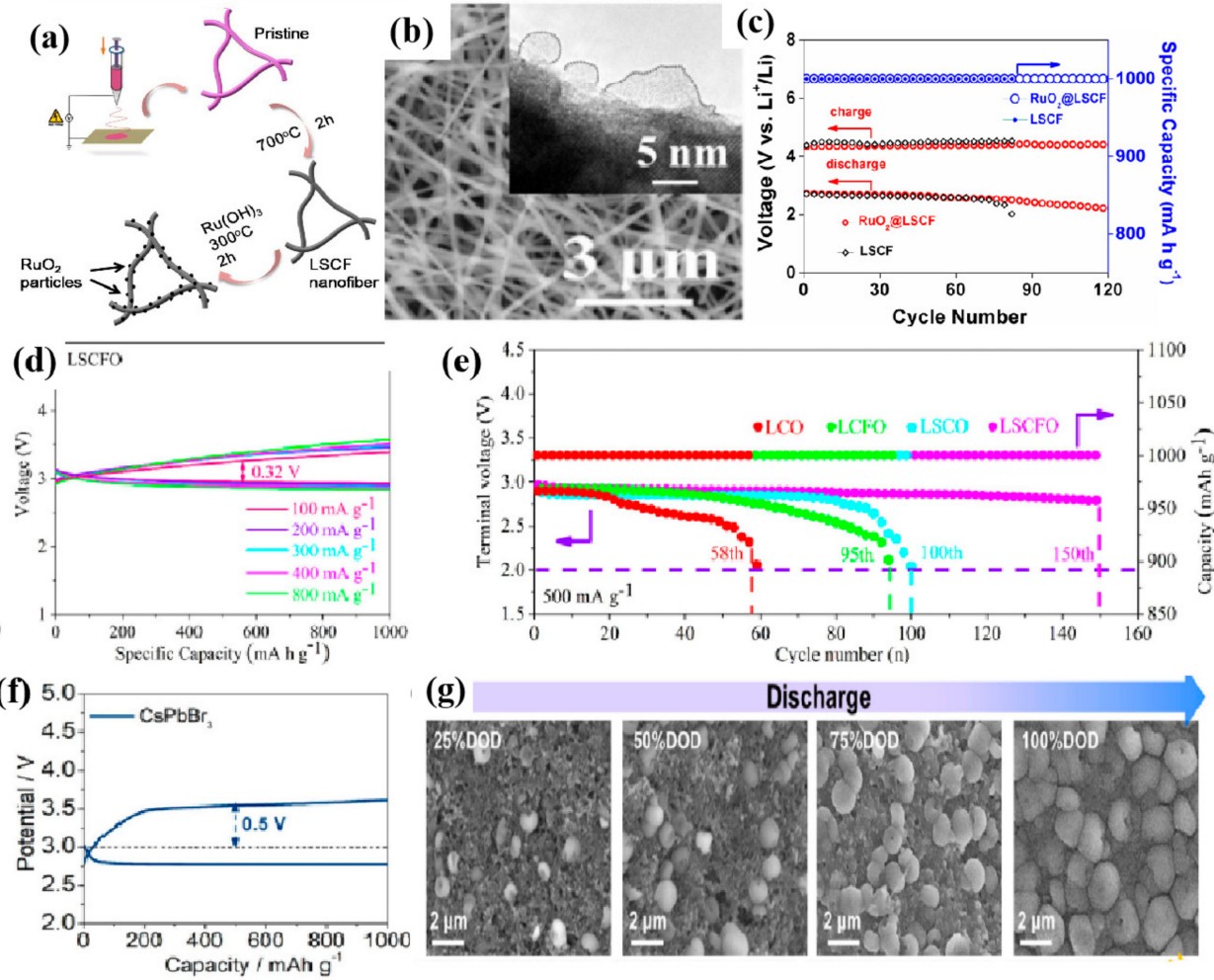

**Figure 12.** (**a,b**) Schematic diagram and SEM/TEM image of simple synthesis of $RuO_2$ nanoparticles@LSCF-NFs. (**c**) Battery discharge/charging terminal voltage $RuO_2$@LSCF -NFS and LSCF-NFs catalysts with a cut-off capacity of 1000 mAhg$^{-1}$ at 100 mAg$^{-1}$. Reprinted with permission from ref. [195]. Copyright 2017 John Wiley and Sons. (**d**) Magnification performance of LABs based on LSCFO electrode. (**e**) Cycle performance of LABs at 500 mA $g^{-1}$ current density and capacity of 1000 mA h $g^{-1}$. Reprinted with permission from ref. [200]. Copyright 2021 American Chemical Society. (**f**) Discharge and charging curves of $CsPbBr_3$ cathode with capacity limited to 1000 mA h $g^{-1}$ and current density of 0.1 A $g^{-1}$. (**g**) Morphological evolution of $CsPbBr_3$ cathode during discharge. Reprinted with permission from ref. [201]. Copyright 2021 American Chemical Society.

*3.4. Other Kind of Transition Metal Compounds*

Except for the metal oxides, the sharpness of the other kind of transition metal compounds used as the cathode catalysts is coming, such as the metal nitride [202–204], metal sulfide [205–210], metal carbide [211–215], metal phosphide [216–219] and metal hydroxide [220–225].

$Co_4N$-decorated CNF ($Co_4N$/CNF) cathode revealed outstanding cycle stability over 177 cycles with a restricted capacity of 500 mA h $g^{-1}$ and a good rate capability with various ampere densities (100, 200, 400, 600, and 800 mA $g^{-1}$) could also be detected with a limited capacity value of 1000 mA h $g^{-1}$ [203]. This outstanding electrochemical performance of the $Co_4N$/CNF could be attributed to sufficient active sites provided by the $Co_4N$ nanorods and the excellent electronic transmission capability of the CNF. Due to the good synergistic effect of uniformed nanoparticles equally distributed on the surface of the porosity N-doped graphene (NG), the $Fe/Fe_3N/Fe_4N$@NG nanocomposites delivered a charming discharge capacity up to 7397.8 mAhg$^{-1}$ at 100 mAg$^{-1}$ (Figure 13a–c) [224]. The cell can be operated over 100 stable loops with the overpotential limited to 2 V, and it can be deliver about 130 cycles with the fixed capacity of 1000 mA h $g^{-1}$ at 100 mA $g^{-1}$.

Dual heteroatom-doped 3D carbon nanofoam-wrapped FeS nanoparticles (FeS-C) at the cathode catalysts of the LABs show a full discharge capacity up to 14,777.5 mA h $g^{-1}$ at 0.1 mA $cm^{-2}$, and it can deliver a stable cycle for 100 cycles with a cut-off capacity of 500 mA h $g^{-1}$ at 0.3 mA $cm^{-2}$ [206]. A well-designed Fe CoP/CC nanowire array was successfully fabricated. Fe CoP/CC-based LABs have excellent oxygen electrode reaction catalytic activity, sufficient $Li_2O_2$ accommodation space, and abundant $O_2$ transfer channels, showing good reversibility and excellent cycle stability (Figure 13d) [226]. An efficient self-supporting $ZnCo_2S_4$ catalyst for nonaqueous LABs was synthesized. The catalyst has a uniform hollow structure. The $ZnCo_2S_4$ cathode demonstrated excellent battery performance. It had a high initial discharge capacity of 9505 mA h $g^{-1}$ at 100 mA $g^{-1}$, a small overpotential of 1.02 V at 100 mA $g^{-1}$, an excellent rate performance of 4988 mA h $g^{-1}$, an extended cycle life of 400 mA $g^{-1}$, and 90 loops at 100 mA $g^{-1}$ [227]. The Bismuth sulfide nanorod array decorated nickel foam (R-$Bi_2S_3$/NF) electrode displayed an energy efficiency of about 78.8% and a cyclability over 140 cycles [228]. Honeycomb $Ni_2P/Ni_{12}P_5$ heterostructures rich in phosphorus vacancy were grown in situ on nickel foam ($Ni_2P/Ni_{12}P_5$@NF), which became a highly efficient Li-O dual-function oxygen electrode. The enhanced catalytic activity of the honeycomb $Ni_2P/Ni_{12}P_5$ heterostructure mainly comes from the synergy of unique electronic recombination interface and abundant phosphorus vacancies. Due to the strong electron coupling effect, the modulated electron density at the active center accelerated the electron transport, which is conducive to regulating the adsorption and activation behavior of oxygen-containing intermediates in the reaction process. In addition, a large number of phosphorus vacancies can not only provide additional active sites but also promote the delocalization of bound electrons around the Ni-P bond to increase the conductivity of effective electron transport and reduce the electrochemical impedance. Therefore, excellent battery performance can be achieved, including an excellent discharge-specific capacity of 13,254.1 mA h $g^{-1}$, a low voltage gap of 0.89 V, and a long loops life of more than 500 h (Figure 13e,f) [229].

The $MoC_{1-x}$/HSC nanocomposites as the cathode of the LABs reveal an overpotential just for 0.6 V at a current density of 100 mA $g^{-1}$ with a restrained capacity of 1000 mA h $g^{-1}$, and it can maintain the capacity fixed over 100 cycles [212]. Nitrogen-doped defective $MoS_2$ catalyst was successfully prepared by hydrothermal method combined with ammonia calcination process, and suggestions to enhance the electrochemical properties of LABs were put forward. The homogeneous N-doped $MoS_2$ catalyst with many defects and porous structure shows excellent catalytic activity, which is conducive to the kinetics of oxygen reaction and the decomposition of discharge products. The LABs composed of the N-$MOS_2$ cathode catalyst showed a high initial specific capacity of 5500 mA h $g^{-1}$ at 100 mA $g^{-1}$ (Figure 13g,h) [230].

The MoP@CC electrode was synthesized and used as the cathode for the LABs by Wei et al. [216]. The battery shows good rate capacity and reduced overpotential, and it can work for 400 cycles under the cut-off capacity of 0.25 mA h $g^{-1}$ at 0.1 mA $g^{-1}$ without obviously fading. The porous cobalt phosphide nanosheets synthesized by Huang et al. exhibited an improved electrochemical performance [217]. The $Co_2P/Ru/N$-doped carbon nanotube hybrids ($Co_2P/Ru/CNT$) reported by Wang et al. performed a lower overpotential of about 0.75 V and cycling stability of more than 185 cycles could be obtained under the controlled capacity of 1000 mA h $g^{-1}$ at 100 mA $g^{-1}$ [219]. An outstanding initial full discharge capacity was detected up to 18,048, 14,217, and 12,800 mA h $g^{-1}$ at 100, 400, and 1000 mA $g^{-1}$, respectively, which suggested an excellent rate capacity.

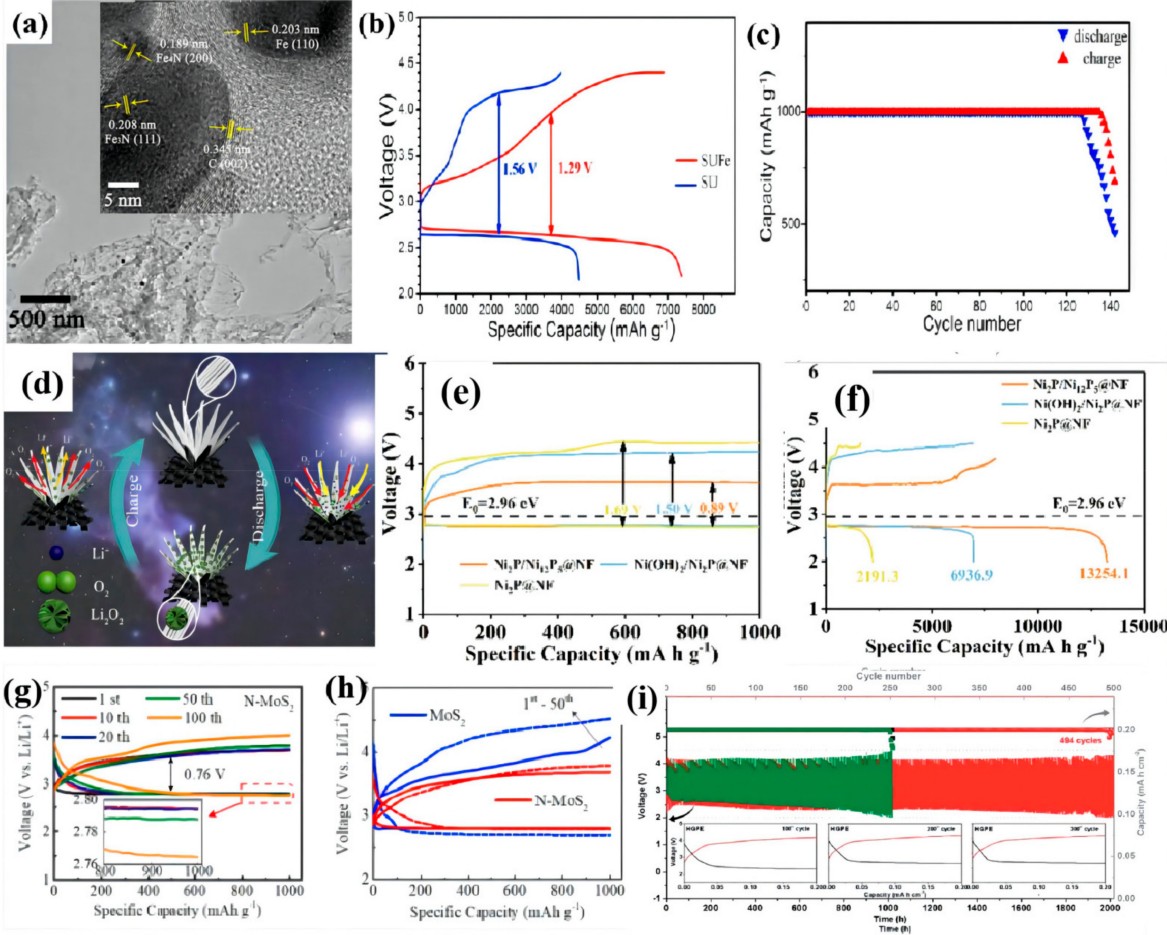

**Figure 13.** (**a**) TEM images of the $Fe/Fe_3N/Fe_4N@NG$ nanocomposites (SUFe), (**b**) The initial discharge and charge curves of the LABs based on SUFe and SU, (**c**) the cycle profiles of the LABs based on SUFe with a fixed capacity of 1000 mA h $g^{-1}$ at 100 mA $g^{-1}$. Reprinted with permission from ref. [224]. Copyright 2018 John Wiley and Sons. (**d**) Reaction mechanism of Fe CoP/CC based LABs. Reprinted with permission from ref. [230]. Copyright 2020 Elsevier. (**e**,**f**) The first discharge/charge curve with a current density of 500 mA $g^{-1}$ and the discharge charge curve with a fixed capacity of 1000 mA h $g^{-1}$ for $Ni_2P/Ni_{12}P_5$ @ NF, $Ni(OH)_2/Ni_2P@NF$ and $Ni_2P@NF$ Electrode. Reprinted with permission from ref. [229]. Copyright 2021 Elsevier. (**g**) Cyclic test of n-$MOS_2$ (**h**) The cyclic voltammetry curve with a scanning rate of 0.02 mV $s^{-1}$ between 2.2 and 4.4 V. Reprinted with permission from ref. [230]. Copyright 2021 Elsevier. (**i**) Electrochemical performance of $Li$-$O_2$ batteries with the HGPE and GPE at 0.1 mA $cm^{-2}$ with a limited capacity of 0.2 mA h $cm^{-2}$. Reprinted with permission from ref. [226]. Copyright 2019 Royal Society of Chemistry.

As a cathode catalyst of the LABs the binary hydroxides of cobalt and iron, exhibit an fairly stable cyclability for 20 cycles at 0.2 mA $g^{-1}$ [220]. NiFe-layered double hydroxide

prepared by a hydrothermal method indicated a good catalytic performance with a full discharge capacity about 3218 mA h g$^{-1}$ at 0.1 mA cm$^{-2}$ [221]. And the reports exhibited that the electrocatalysts performance had been improved greatly by the MnOOH [222] and FeOOH [223].

Guo et al. reported that the γ-MnOOH nanorods mixed with the carbon composite as the catalyst exhibited the original full discharge capacity higher, up to 9400 mA h g$^{-1}$ at 50 mA g$^{-1}$, and the battery operated for 100 cycles stability at 200 mA g$^{-1}$ [222]. Recently, benefiting from the improved active surface area and reduced charge-transfer resistance, the RuO$_2$ decorated MnOOH/C as the air electrode of the LABs displayed a prolonged cycle performance up to 200 cycles with a limited capacity of 1000 mA h g$^{-1}$ at 0.2 mA cm$^{-2}$. Zhao and his coworkers used the MnOOH nanowires to form the MnOOH @Al$_2$O$_3$ films as the skeleton to prepare the nanowire-film-reinforced hybrid GPE (HGPE) acting as the air cathode of the LABs (Figure 13i) [226]. With the merits of the lightweight, coherent, and highly porous, the cell with the HGPE cathode demonstrated superior cycle stability, up to 494 cycles without obviously capacity decay, with a cut-off capacity of 0.2 mA h cm$^{-2}$ at 0.1 mA cm$^{-2}$.

β-FeOOH decorated porous carbon aerogels had been reported by Chen et al. [223]. When the composites are used as the air cathode of the LABs, a pristine full discharge capacity higher up to 10,230 mA h g$^{-1}$ could be detected at 0.1 mA cm$^{-2}$ and a superior C-rate performance could also be obtained. The carbon ribbons anchored with the FeOOH nanocubes were operated as the catalysts of the LABs by Lin et al. [231]. Due to the synergistic effect, an initial full discharge capacity of 14,816 mAhg$^{-1}$ can be demonstrated at 200 mA g$^{-1}$, and long-term stability of 67 cycles over 400 h could be tested under a controlled capacity of 600 mA h g$^{-1}$ at 200 mA g$^{-1}$. Ni-Fe LDHs-V Ni was designed as cathode material by hydrothermal method, which was used as cathode catalyst of LABs. Ni-Fe LDHs-V Ni can show a low charging overpotential (0.48 V). The enhancement performance of Ni-Fe LDHs-V Ni is mainly due to the rich Ni vacancies in the positive electrode of Ni-Fe LDHs-V Ni, which improves the adsorption energy of intermediates. According to DFT calculation, the Ni vacancy in Ni-Fe LDHs-V Ni can enhance the adsorption capacity of LiO$_2$, resulting in low charging voltage through the catalysis of Ni vacancy [232].

These results suggested that metal hydroxide is also a promising LAB catalyst with some decorating or modifying to tune the catalytic property. Moreover, more emerging efforts and in-depth studies should be devoted to broadening the research scope of metal hydroxide.

## 4. Conclusions, Progress and Outlook

In this review, the carbon materials, transition metal chalcogenides, and their composites, which acted as the bifunctional cathode catalysts of the LABs, are reviewed comprehensively, including the material selection, synthesis method, structural/morphology characterization, and electric catalytic performance. Compared with single species catalysts, catalysts with multielement compounds or nanocomposites contain the synergistic effects from the different components with manipulation of the complex and ever-changing micro/nanostructure to improve the ORR and OER performances of the LABs air cathode catalysts. In conclusion, the integration of various functional and multidimensional materials is a competitive way to achieve high performance. Hence, to design advanced bifunctional composite catalysts, selecting the proper original component is crucial because the properties of the components materials affect the transmission rates of the electrons/ions/oxygen gas and even the distribution of the active catalytic site. Furthermore, the interaction between each composition or between the compositions catalysts with the substrate material displayed another important character in determining the ORR/OER performance and the stability/durability of the air cathode catalysts of the LABs. The morphology plays a crucial role in influencing the full electrocatalytic performance of the electrode, such as the porous-structured materials having a higher surface area which provides more active sites and more places for the Li$_2$O$_2$ nucleation/decomposition. Shorter conducting paths for the

ions/electron's transportation can be acquired with the porous structure. For example, the graphene/graphene-liked material composited with the metal oxide as the bifunctional catalysts of the LABs, the carbon corrosion/oxides had been overcome with the synergistic interactions that were induced by the composited metal oxides to form the carbon-based metal oxides nanocomposites. The ORR/OER performance of the interacting components was improved with the increasing interactions between carbon and metal oxides. In addition, as the electrode of the LABs, its electrocatalytic activity, and cycling performance were also improved.

As mentioned above, the graphene-like materials and the polymetallic compounds exhibit the outstanding bifunctional catalytic activity of ORR and OER, and the polymetallic compounds/carbon material nanocomposites enhance the full electrochemical performance of the air cathode. In addition, the synergistic effect improves the performance obviously with the integration of multifarious functional and multidimensional materials and the strong interaction between individual materials.

In the past few years, despite many efforts devoted to the development of the composited bifunctional catalysts for the LABs, there are still some major technical bottlenecks that limit the commercial application of catalysts. The main limitations of developing catalysts for LABs are shown below: (i) the insufficient electrocatalytic performance for the major electrocatalytic reaction. As mentioned above, there are two main electrode reactions in the LABs, the ORR and OER, with limited electrocatalytic activity. The ORR/OER has been sluggish, which leads to a lower energy density and lower coulomb efficiency of the LABs and (ii) insufficient stability/durability of cathode catalysts. The catalyst surface may be damaged/corroded or covered by indecomposable $Li_2O_2$ and $Li_2CO_3$, resulting in the reduction of active centers and structural damage, limited space for the nucleation/decomposition of reaction products, and the degradation of the LABs performance; (iii) unoptimized designing strategies for catalyst and electrode. Exposing more active sites, increasing the specific surface area, enhancing the interactions of each component, and the electronic transmission capability of the integrated components are the major ways to enhance the full electrocatalytic performance. However, the optimized comprehensive arrangements and optimized synthesis strategy has a long way to go; (iv) insufficient fundamental understanding of catalyst interaction. It is well known that the ORR/OER is a complex and sluggish process, and the chemical processes, thermodynamics, and kinetics are still not clear enough. Much effort is still needed to explore the fundamental theory, as well as the design and fabrication principles of the electrode for LABs. In summary, bifunctional catalysts for air cathodes in Li-ion batteries are still in their immature stage and thus require extensive optimization and transition from concept stage to large-scale production in a cost-effective manner. Meanwhile, understanding of basic theory will be extremely valuable for material selection and optimization of catalyst/electrode designs.

In order to conquer above-mentioned technical defects, several comments on directions for future research could be suggested for the fabrication and performance of the next generation bifunctional composites catalysts for the LABs air cathodes.

(I). Enhance the activity and the stability/durability of the cathode catalysts. Some new concepts on the design, synthesis, and fabrication for the catalysts/electrode should be implemented, such as designing some special structure to expose more active sites, synthesizing the porous structured material to obtain larger surface area, and optimizing the pore size and distribution to get the performance. With the morphology and micro/nanostructure-controlled synthesis strategies, the electrocatalytic performance can be improved.

(II). A thorough and systematic study should be operated to understand the electrocatalytic mechanism of the bifunctional composite catalysts for the air cathode of the LABs. The detailed experimental results should be used to verify the correctness of the theoretical calculation and correct its deviation. Here, the experimental results should be combined with molecular/atomic dynamics models and thermodynamic models, demonstrating the tight connection between the catalytic ORR/OER mechanism and

the electronic structure/composition of the catalyst. With the right mechanism, the design/synthesis strategy of electrode catalysts could be the most reasonable, and the performance of ORR/OER for the LABs should be improved greatly.

(III). Introduce a new, simple, and highly efficient synthesis method with low-cost green materials to fabricate the catalyst materials with high-performing and cost-effective bifunctional composite catalysts as the cathode of the LABs. The multi-compound/doping can possess synergistic activity interactions, enhancing the electrochemical catalytic activity and loops-stability of the composited catalysts.

We believe that with the further development of material preparation technology, rational micro/nanostructure, and multi-component design will inevitably improve the overall performance of bifunctional composite catalysts for LABs air cathodes. It is hoped that this major advance can provide some constructive comments for researchers in related fields and attract more research attention to this field.

**Author Contributions:** Conceptualization, G.Y.; methodology, Z.H. and Y.X.; software, Z.H. and Y.X.; validation, Z.H. and Y.X.; formal analysis, Y.X.; resources, Y.X. and Z.H.; data curation, Z.H., Y.X., T.Y. and Y.W.; writing—original draft preparation, G.Y., Z.H. and Y.X.; writing—review and editing, G.Y. and Y.X.; visualization, Z.H.; supervision, G.Y.; project administration, G.Y.; funding acquisition, G.Y. All authors have read and agreed to the published version of the manuscript.

**Funding:** The authors gratefully acknowledge the financial support from the National Natural Science Foundation of China (grant nos. 51971184 and 51931006); the Natural Science Foundation of Fujian Province of China (no. 2021J01043), the Fundamental Research Funds for the Central Universities of China (Xiamen University: no. 20720200068), and the "DoubleFirst Class" Foundation of Materials Intelligent Manufacturing Discipline of Xiamen University.

**Institutional Review Board Statement:** Not applicable.

**Informed Consent Statement:** Not applicable.

**Data Availability Statement:** Data sharing is not applicable to this article.

**Conflicts of Interest:** The authors declare no conflict of interest.

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
