# Peer review of "Bifunctional Electrocatalysts Materials for Non-Aqueous Li–Air Batteries"

_coatings, doi:10.3390/coatings12081227_

Round 1

Reviewer 1 Report

This manuscript is about a review on bifunctional electrocatalyst materials for Li-air batteries. This manuscript can be accepted after minor correction. 

1.  I would suggest to include “materials” in the title. Also, it should be “non-aqueous”.

2.     Please provide the architecture of Li-air battery for better understanding.

3.     Standardize the term of Li-air batteries (either LABs or Li-O2 batteries).

4.     Figures provided in the manuscript have not been mentioned in the text. Therefore, it does not give any meaning to the manuscript. 

5.   Why transition metal chalcogenides? Normally chalcogenides refer to chalcogen elements like S, Se and Te even though oxygen is in the same group of them. Section 3.3 can be renamed as Transitional Metal Oxides and section 3.3.3 to be renumbered to 3.4. 

Author Response

The responses to the editor’s and reviewers’ comments and the revision details

Title: Bifunctional electrocatalysts materials for non-aqueous Li–air batteries

Journal: Coatings

We have carefully revised our manuscript by considering the reviewers’ comments. All the changes are highlighted with yellow background in the revised manuscript. The responses to the editor’s and reviewers’ comments and the revision details are as follows:

 Responds to the reviewers’ comments:

Reviewer #1-Comment 1:  I would suggest to include “materials” in the title. Also, it should be “non-aqueous”.

Response: Thanks for your kind suggestion. We have changed the title to Bifunctional electrocatalysts materials for non-aqueous Li-air batteries

Reviewer #1-Comment 2:  Please provide the architecture of Li-air battery for better understanding.

Response: Thanks for your kind suggestion. We have added fig. 1 for understanding.

Fig. 1. The architecture of LABs. Source: Girishkumar et al. 2014

Reviewer #1-Comment 3: Standardize the term of Li-air batteries (either LABs or Li-O2 batteries).

Response:  Thanks for your kind suggestion. We have unified this statement as LABs.

Reviewer #1-Comment 4: Figures provided in the manuscript have not been mentioned in the text. Therefore, it does not give any meaning to the manuscript.

Response:  Thanks for your kind suggestion. All the figures are mentioned in the text now.

Reviewer #1-Comment 5: Why transition metal chalcogenides? Normally chalcogenides refer to chalcogen elements like S, Se and Te even though oxygen is in the same group of them. Section 3.3 can be renamed as Transitional Metal Oxides and section 3.3.3 to be renumbered to 3.4. 

Response:  Thanks for your kind suggestion. We have revised the title of section as you point out.

Section 3.3 Transition Metal Oxides

Section 3.4 Other kind of transition metal compounds

Reviewer 2 Report

The reviewed work is an extensive review of cathodic materials for lithium-air batteries (LABs). While it is well organized and contains all major ORR carbon and non-carbon catalysts investigated for use in LABs, the manuscript requires a very serious correction of grammar and style as it is not suitable for publication in the present form.

Author Response

Reviewer #2: The reviewed work is an extensive review of cathodic materials for lithium-air batteries (LABs). While it is well organized and contains all major ORR carbon and non-carbon catalysts investigated for use in LABs, the manuscript requires a very serious correction of grammar and style as it is not suitable for publication in the present form.

Response:  Thanks for your kind suggestion. We have revised the WHOLE manuscript carefully and tried to avoid any grammar or syntax error. In addition, we have asked several colleagues who are skilled authors of English language papers to check the English. We believe that the language is now acceptable for the review process.

Round 2

Reviewer 2 Report

This is a revision of a previously submitted manuscript. Although the grammar and style have been improved significantly, there are still present sporadic grammar glitches here and there, e.g., "circulations" on p. 13 should be changed to "cycles". 

Author Response

The responses to the editor’s and reviewers’ comments and the revision details

Title: Bifunctional electrocatalysts materials for non-aqueous Li–air batteries

Journal: Coatings

We have carefully revised our manuscript by considering the reviewers’ comments. We are using “track changes” model in MS word and also marked the changes with red color in the revised manuscript. The responses to the editor’s and reviewers’ comments and the revision details are as follows:

Responds to the reviewers’ comments:

Reviewer:  This is a revision of a previously submitted manuscript. Although the grammar and style have been improved significantly, there are still present sporadic grammar glitches here and there, e.g., "circulations" on p. 13 should be changed to "cycles".

Response: Thanks for your kind suggestions. We have carefully checked our manuscript several times and made some revisions. We hope there are less mistakes this time. We are using “track changes” model in MS word and also marked the changes with red color.

Page 1: “cathode-catalyst” to “cathode-catalysts”; “low price” to “lower price”; “methods” to “strategies”.

Page 3: “change” to “changes”

Page 4: deleted “technologies”

Page 6: Inserted “to”

Page 7: “lower” to “low”

Page 8: deleted “Through research,”; “catalyst” to “catalysts”

Page 12: “detective” to “defective”; “shows” to “showed”

Page 13: “circulations” to “cycles”

Page 16: Inserted “carbon atoms”

Page 24: Inserted “to have”

Page 27: “circulations” to “cycles”

Page 29: Deleted “be” and “were”; “includes” to “including”; “show” to “showed”

Page 30: “those” to “these”

Page 31: “graded” to “hierarchical”

Page 35: “nanopartical” to “α-MnO2 nanowire”

Page 39: Deleted “with other”

Page 40: Deleted “were”

Page 49: “enhance” to “enhancement”

Page 50: “time” to “times”

Page 56: “were” to “was”

Page 57: “prepared” to “prepare”

Page 59: “select” to “selecting”

Page 60: “dived” to “devot
